# Mapping the degradation pathway of a disease-linked aspartoacylase variant

**Sarah K. Gersing**[1]*, **Yong Wang**[1], **Martin Grønbæk-Thygesen**[1], **Caroline Kampmeyer**[1], **Lene Clausen**[1], **Martin Willemoës**[1], **Claes Andréasson**[2], **Amelie Stein**[1]*, **Kresten Lindorff-Larsen**[1]*, **Rasmus Hartmann-Petersen**[1]*

1 The Linderstrøm-Lang Centre for Protein Science, Department of Biology, University of Copenhagen, Copenhagen, Denmark, 2 Department of Molecular Biosciences, The Wenner-Gren Institute, Stockholm University, Stockholm, Sweden

* sarah.gersing@bio.ku.dk (SKG); amelie.stein@bio.ku.dk (AS); lindorff@bio.ku.dk (KL-L); rhpetersen@bio.ku.dk (RH-P)

**Data Availability Statement:** All relevant data are within the manuscript and its Supporting Information files.

## Abstract

Canavan disease is a severe progressive neurodegenerative disorder that is characterized by swelling and spongy degeneration of brain white matter. The disease is genetically linked to polymorphisms in the aspartoacylase (*ASPA*) gene, including the substitution C152W. ASPA C152W is associated with greatly reduced protein levels in cells, yet biophysical experiments suggest a wild-type like thermal stability. Here, we use ASPA C152W as a model to investigate the degradation pathway of a disease-causing protein variant. When we expressed ASPA C152W in *Saccharomyces cerevisiae*, we found a decreased steady state compared to wild-type ASPA as a result of increased proteasomal degradation. However, molecular dynamics simulations of ASPA C152W did not substantially deviate from wild-type ASPA, indicating that the native state is structurally preserved. Instead, we suggest that the C152W substitution interferes with the *de novo* folding pathway resulting in increased proteasomal degradation before reaching its stable conformation. Systematic mapping of the protein quality control components acting on misfolded and aggregation-prone species of C152W, revealed that the degradation is highly dependent on the molecular chaperone Hsp70, its co-chaperone Hsp110 as well as several quality control E3 ubiquitin-protein ligases, including Ubr1. In addition, the disaggregase Hsp104 facilitated refolding of aggregated ASPA C152W, while Cdc48 mediated degradation of insoluble ASPA protein. In human cells, ASPA C152W displayed increased proteasomal turnover that was similarly dependent on Hsp70 and Hsp110. Our findings underscore the use of yeast to determine the protein quality control components involved in the degradation of human pathogenic variants in order to identify potential therapeutic targets.

## Author summary

Canavan disease is a fatal neurodegenerative disorder which is genetically linked to polymorphisms in the aspartoacylase (*ASPA*) gene. Although the molecular mechanism of most disease-causing substitutions remains to be examined, some variants have been

**Funding:** This work was supported by the Novo Nordisk Foundation (https:// novonordiskfonden. dk) challenge programme PRISM (to K.L.L., A.S. & R.H.P.) and NNF18OC0052441 (to R.H.P.), the Lundbeck Foundation (https://www. lundbeckfonden.com) R249-2017-510 (to L.C. & R.H.P.) and R272-2017-452 and R209-2015-3283 (to A.S.), and Danish Council for Independent Research (Natur og Univers, Det Frie Forskningsråd) (https://dff.dk) 7014-00039B (to R. H.P.). The funders had no role in study design, data collection and analysis, decision to publish, or preparation of the manuscript.

**Competing interests:** The authors have declared that no competing interests exist.

suggested to cause the loss-of-function phenotype by perturbing the structural stability of ASPA. So far the cellular fate of these variants have not been examined. Here we examine the stability and degradation pathways of the disease-causing ASPA variant C152W. In yeast cells, ASPA C152W showed decreased steady-state protein levels as a result of increased proteasomal turnover. Our molecular dynamics simulations showed that the C152W substitution did not globally perturb the native structure of ASPA. Instead we propose that ASPA C152W is targeted by the protein quality control system during *de novo* folding. Specifically, we found that the molecular chaperone Hsp70, its co-chaperone Hsp110, and the E3 ubiquitin-protein ligase Ubr1 promote degradation of ASPA C152W. When we expressed ASPA C152W in cultured human cells, we found that Hsp70 and Hsp110 similarly mediated degradation. Therefore, we propose that Hsp110 should be further examined as a potential therapeutic target in Canavan disease and other protein misfolding diseases.

## Introduction

Canavan disease (OMIM: 271900) is a hereditary leukodystrophy resulting in severe and progressive neurodegeneration. The disorder is characterized by swelling and spongy degeneration of brain white matter. Clinical signs often appear in the first few months of life and include poor head control, macrocephaly, hypertonia, and severe developmental delay. Death typically occurs in early childhood. Loss of enzymatic activity in aspartoacylase (ASPA) is tightly linked to Canavan disease. The human *ASPA* gene is located on chromosome 17 where it spans 30 kilobases consisting of five introns and six exons [1]. Within the *ASPA* gene, more than 50 variants have been linked to Canavan disease, primarily consisting of single nucleotide substitutions [2]. Although the disease is most prevalent among people of Ashkenazi Jewish origin [3], it has been reported in populations worldwide [2,4].

ASPA is predominantly expressed in oligodendrocytes where it functions as a homodimer and is required for the hydrolysis of *N*-acetylaspartate (NAA) to acetate and aspartate [5–7]. Lack of ASPA activity results in the accumulation of its substrate, NAA. Consequently high levels of NAA in urine can be used as a biochemical marker for Canavan disease [8]. However, next-generation sequencing technologies could potentially improve diagnosis and provide a more accurate prognosis for patients [9]. The mechanism linking loss of ASPA function to the phenotypes associated with Canavan disease is still debated. One hypothesis is that NAA functions as a brain osmolyte to remove neuronal metabolic water [10]. Others propose that NAA functions as an acetate carrier, providing oligodendrocytes with acetate for lipid synthesis and myelination [8,11]. There is currently no approved therapy for Canavan disease, although enzyme replacement in a mouse model and a clinical trial using gene therapy both resulted in reduced brain NAA levels [12,13]. In addition, a recent study demonstrated rescue of several pathological features associated with Canavan disease in a mouse model using human induced pluripotent stem cell (iPSC)-based cell therapy [14].

On the molecular level, some disease-linked ASPA variants have been found to trigger the loss-of-function phenotype by perturbing the active site or protein stability [15,16]. However, the majority of disease-linked ASPA variants remain to be examined. In addition, the fate of destabilized ASPA variants in the cell is unknown. Our focus in the present study is on the C152W variant, which has been associated with loss of ASPA activity *in vitro* and accordingly Canavan disease [17]. The C152 residue has been hypothesized to participate in an intramolecular disulphide bond with C124 [15,18]. However, this remains to be established, and crystal

structures of ASPA do not show a disulphide bond [19,20]. The C152W variant is associated with greatly reduced protein levels [15], although biophysical experiments suggest a wild-type like thermal stability [21]. In this study ASPA C152W is used as a model to study cellular protein degradation of disease-causing variants showing reduced protein levels.

In general, disease-causing missense variants can destabilize a protein enough to cause protein unfolding, which in turn results in loss of protein function [22–24]. In addition, even a subtle destabilization might cause transient and/or local protein unfolding events or delay *de novo* folding, leading to increased exposure of regions of the protein that are recognized by the cellular protein quality control (PQC) system, which in turn may target the protein for degradation. Protein regions that function as degradation signals are termed degrons. For PQC degrons, these mainly appear to be buried stretches of hydrophobic amino acid residues [25], however, despite substantial research the exact patterns have not been elucidated [26–28]. The key players in recognition of misfolded and unfolded proteins are molecular chaperones, which promote the folding of nascent and damaged proteins. If folding fails, the chaperones cooperate with the ubiquitin-proteasome system (UPS) to ensure that non-native proteins are cleared [29]. The PQC system thus ensures that misfolded and/or aggregated proteins, which may be toxic, do not accumulate in the cell. However, since the PQC system does not probe for function and likely recognizes characteristics specific for unfolded proteins, such as exposed hydrophobic regions, protein variants that cause exposure of these regions might lead to degradation of functional proteins [30,31].

Consequently, it might be possible to uncouple loss of protein stability from loss of function [32], if components of the PQC system required for a protein's degradation can be identified and targeted. Since loss of protein stability is proposed to be a central cause of human diseases [33–36], this strategy holds wide-ranging therapeutic potential. Therefore it is of major relevance to further characterize the components of the PQC system that are required for the degradation of misfolded proteins.

Previously, proteasome inhibition has been used to rescue the function of deleterious protein variants [37]. However, since the proteasome is essential for cell viability, targeting upstream PQC components involved in the selection of individual substrates will likely be more specific and therefore more attractive for therapeutic purposes [38]. Components of the PQC system that could potentially be targeted include E3 ubiquitin-protein ligases [37], which provide the substrate specificity in the process of ubiquitination. Due to the abundance of diverse E3 ligases encoded in the human genome [39], these are likely to be fairly specific targets. However, E3s show redundancy regarding substrate degradation [37,40,41], making it difficult to single out individual E3s and hence efficiently prevent degradation of the target protein. In addition to E3s, molecular chaperones could be potential therapeutic targets due to their role as substrate recognition factors in the UPS [42–46].

Here, using the yeast *Saccharomyces cerevisiae* as a model organism, we show that ASPA C152W displays a reduced cellular protein level as the variant is rapidly degraded via the UPS. Our molecular dynamics simulations suggest that the C152W substitution does not globally perturb the native structure of ASPA. Instead, the C152W substitution might impede *de novo* folding of ASPA leading to increased recognition and degradation by the PQC system. Proteasomal degradation of ASPA C152W depends on the molecular chaperones Hsp70 and Hsp110, and multiple E3 ubiquitin-protein ligases, including Ubr1. Importantly, the identified degradation pathway is conserved to human cells, thus emphasising that budding yeast is an excellent system for pinpointing PQC components required for degradation of a specific target, such as disease-causing human protein variants. We propose that Hsp110 should be further investigated as a potential therapeutic target to stabilize some ASPA variants and possibly other disease-linked missense protein variants. However, low abundance protein variants,

including ASPA C152W, might also affect enzyme catalysis in addition to stability and/or folding [32], and therefore targeting molecular chaperones is not a universal therapeutic strategy.

## Results

### Proteasomal degradation and aggregation of ASPA C152W

To learn more about the stability and degradation pathways of ASPA C152W, we employed a reporter-based system in *Saccharomyces cerevisiae* [27,47]. In this system, the human *ASPA* cDNA was fused to sequences encoding the Ura3-HA-GFP reporter and expression was driven by the copper-titratable *CUP1* promoter [48] (Fig 1A). Consequently, on medium lacking uracil the growth of *ura3* yeast cells carrying this construct depends on the stability of the Ura3-HA-GFP-ASPA(C152W) fusion protein. Leaky expression from the *CUP1* promoter in absence of copper was used for growth experiments unless otherwise indicated.

First, the reporter system was tested by examining the growth of yeast cells expressing either wild-type ASPA or the C152W variant. While control cells that expressed the wild-type ASPA fusion grow robustly on liquid or solid medium lacking uracil, growth was severely reduced by the C152W variant (Fig 1B and 1C). The effect was not due to toxicity of the C152W variant since growth was comparable when the medium was supplemented with uracil (Fig 1C). Instead, Western blotting revealed that the protein level of the C152W variant was reduced compared to wild-type ASPA, thus limiting the amount of Ura3 reporter expression (Fig 1D).

To test if the reduced steady-state level of ASPA C152W was the result of increased degradation, the levels of ASPA were determined after translation was inhibited by cycloheximide (CHX). Following one hour of translation arrest there was a greater reduction in the level of ASPA C152W compared to wild-type ASPA (Fig 2A), suggesting that the decreased steady state of ASPA C152W is a result of increased degradation. Next, we examined the involvement of the 26S proteasome in mediating ASPA degradation. Yeast cells expressing wild-type ASPA or the C152W variant were grown on medium lacking uracil in the absence or presence of the proteasome inhibitor, bortezomib (BZ). Proteasomal inhibition removed the growth defect of the C152W variant (Fig 2B). Consistent with the reporter growth readout, the protein level of ASPA C152W was restored upon proteasomal inhibition (Fig 2C). Thus, the ASPA C152W variant exhibits accelerated proteasome-dependent degradation.

To assess if ASPA may be rendered insoluble by the C152W substitution, we next examined the solubility of the two ASPA variants by centrifugation. While wild-type ASPA was mainly soluble, the C152W variant, in contrast, demonstrated decreased solubility (Fig 2D). When the yeast cells were examined by fluorescence microscopy, both variants formed aggregates (Fig 2E and 2G). However, there were significantly more of the aggregate-containing cells expressing ASPA C152W that contained two or three aggregates compared to cells expressing wild-type ASPA (Fig 2H). Consistently, a higher percentage of the total GFP signal was found in aggregates in cells expressing ASPA C152W compared to wild-type (S1A Fig). The aggregates formed by ASPA C152W are unlikely to contribute to the pathogenicity of this variant, as they did not display cellular toxicity (S1B and S1C Fig). When we further characterized the aggregates formed by ASPA C152W, they consistently co-localized with Hsp104 (Fig 2F). However, they did not appear to accumulate in the insoluble protein deposit (IPOD) nor the juxtanuclear quality control (JUNQ) compartment, since we only rarely detected co-localization with Atg8 (S1D Fig) and no consistent association with the nucleus was observed [49]. Instead, the ASPA aggregates might be stress foci or inclusion structures formed prior to establishment of the JUNQ compartment.

Although, previous studies on ASPA C152W have failed to show any residual enzyme activity for this variant when assayed as purified protein or in cell extracts [15,17,21], we attempted

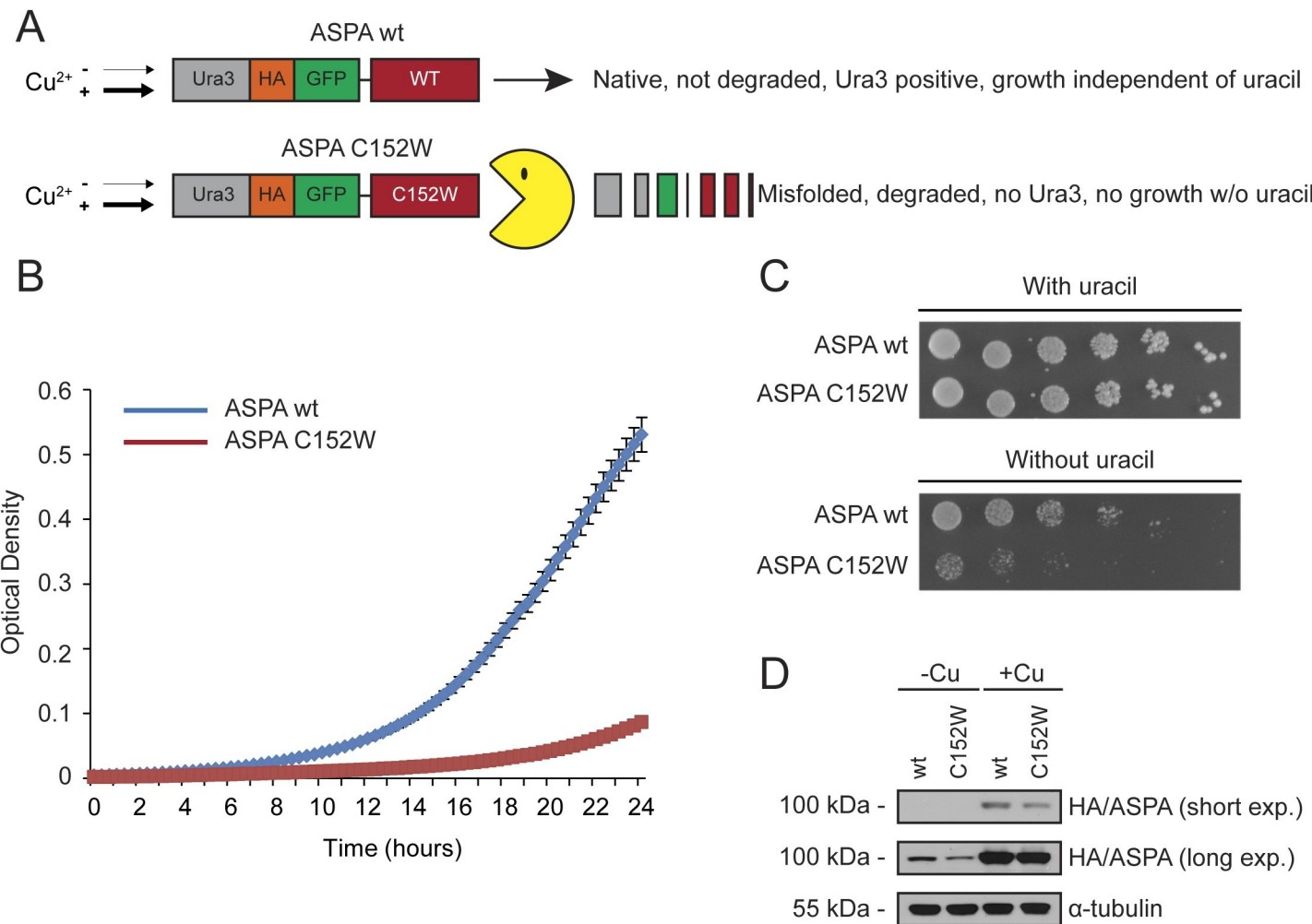

**Fig 1. A growth-based yeast reporter system for protein abundance.** (A) Illustration of the reporter system coupling ASPA steady-state levels to yeast growth. Wild-type ASPA and the C152W variant are expressed from the *CUP1* promoter with an N-terminal fusion construct consisting of Ura3-HA-GFP. When yeast cells express these constructs, their growth in medium lacking uracil depends on the level of the ASPA fusion protein. (B) The growth of wild-type yeast cells expressing the indicated ASPA fusion protein in liquid medium lacking uracil. The optical density was measured at 600 nm. Error bars represent the standard deviation (n = 4). (C) Wild-type yeast cells expressing the indicated ASPA construct grown on solid medium with or without uracil. (D) Western blot of ASPA protein levels in wild-type yeast expressing the indicated ASPA fusion construct. The ASPA fusion proteins are under the control of the *CUP1* promoter, which was either uninduced (-Cu) or induced (+Cu) with copper as shown.

to measure activity by coupling aspartate produced by ASPA to deamination and release of fumarate by *E. coli* aspartase (AspA), as described before [7]. The helper enzyme, AspA, from *E. coli* was produced with a C-terminal 6His-tag and purified (S2A Fig). However, even when we added an N-terminal GST-tag in an attempt to increase solubility, ASPA C152W was not produced in significant amounts (S2B Fig). In addition, the reported very modest enzyme activity ($k_{cat} \sim 0.08$ s$^{-1}$ for *N*-Acetyl-L-aspartate and $k_{cat} \sim 0.62$ s$^{-1}$ for *N*-Chloroacetyl-L-aspartate) for wild-type ASPA means that concentrated ASPA preparations are required [7]. Unfortunately, similar to the previous reports, we were unable to detect any enzyme activity. We therefore instead tested if we could generate an ASPA-based yeast growth assay, by exchanging the carbon or nitrogen source for NAA. However, as we did not observe any ASPA-dependent effects on cell growth (S2C Fig) we did not pursue this further.

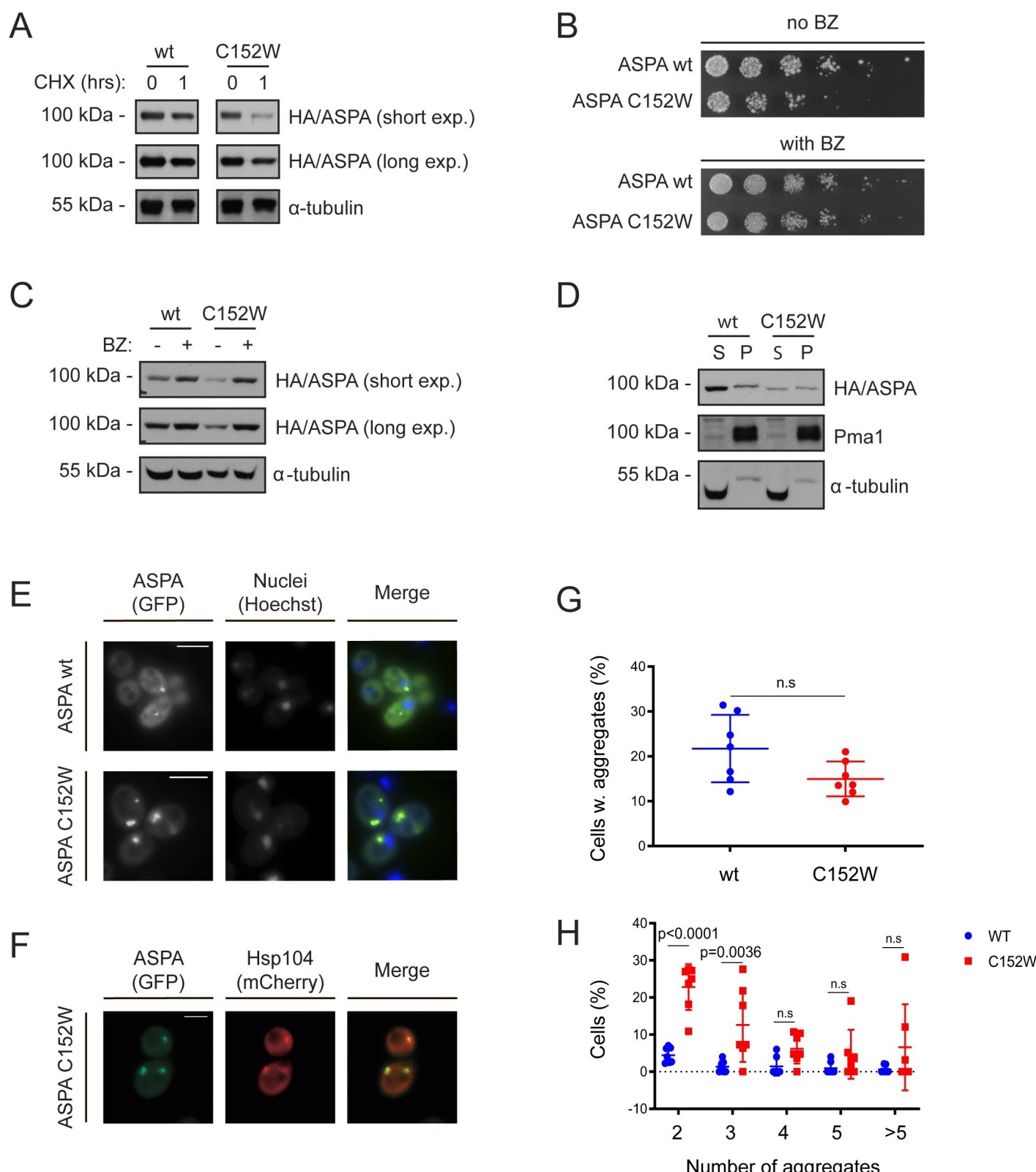

**Fig 2. The ASPA C152W variant is an unstable proteasome target which forms multiple aggregates.** (A) Western blot of ASPA protein levels in wild-type yeast expressing the indicated ASPA variant and treated with the translation inhibitor cycloheximide (CHX) for 1 hour (1) or, as a control, with the solvent DMSO (0). Yeast cells were cultured with 0.1 mM CuSO₄. Blotting for α-tubulin was included as a loading control. (B) Growth of wild-type yeast cells expressing the indicated ASPA construct on solid medium without uracil either in absence (no BZ) or in presence (with BZ) of the proteasome inhibitor bortezomib (BZ). (C) Western blot of ASPA protein levels in wild-type yeast expressing the indicated ASPA variant in presence of 0.1 mM CuSO₄ and treated with DMSO (-) or bortezomib (BZ, +) for

three hours. Tubulin was included as a loading control. (D) Western blot of soluble (S) and insoluble (P) ASPA protein levels in wild-type yeast expressing the indicated construct in presence of 0.1 mM CuSO$_4$. Pma1 and tubulin serve as loading controls for the insoluble and soluble fraction, respectively. (E) Representative fluorescence microscopy images of wild-type yeast cells expressing the indicated ASPA variant. Nuclei were stained with Hoechst. Scale bars are 4 μm. (F) Representative fluorescence microscopy images of wild-type yeast expressing GFP-ASPA C152W and Hsp104-mCherry. ASPA aggregates consistently co-localized with Hsp104, as would be expected for both the JUNQ and the IPOD [49]. Scale bar is 4 μm. (G) Cells expressing GFP and containing aggregates were manually quantified in wild-type yeast expressing wild-type ASPA or the C152W variant using fluorescence microscopy. The frequency of cells containing aggregates in independent experiments are represented as data points (minimum of 282 GFP-expressing cells, n = 7). We used GraphPad Prism to analyse data using an unpaired t-test. Means and standard deviations are shown. (H) Cells containing aggregates and the number of aggregates were quantified in wild-type yeast cells expressing wild-type ASPA or C152W using fluorescence microscopy. The frequency of each type of observation in independent experiments are shown as separate data points (minimum of 282 GFP-expressing cells, n = 7). We used GraphPad Prism to compare wild-type ASPA and C152W aggregates using a two-way ANOVA and Sidak's multiple comparisons test. Means and standard deviations are shown. Three experiments were used for statistics comparing aggregates in wild-type yeast as well as in the *hsp104Δ* strain.

## Exposure of a degron as a possible mechanism for increased turnover of ASPA C152W

Previous computational studies of ASPA C152W have shown that this variant is likely to be destabilized [16] and consistently it exhibits a lower expression level *in vivo* [15]. Somewhat perplexingly, thermal unfolding experiments find the C152W variant to be stable, albeit with very low activity [21]. We revisited the computational analyses of ASPA C152W stability by using molecular dynamics (MD) simulations starting from the crystal structure of ASPA (Fig 3A, visualized with PyMOL [50] and VMD [51]) to examine whether we could detect local unfolding events that might be affected by C152W, and thus performed microsecond-time-scale simulations of the wild type and variant proteins. Our simulations at 310K showed both the wild type and C152W to be stable proteins with conformational fluctuations mostly located in loop regions (Fig 3B). Notably, we did not observe any substantial differences between the two variants. Such atomistic simulations are naturally limited in the time scales they can probe, and we thus repeated these simulations at higher temperatures with the aim to examine whether more subtle differences could be observed. We find that the protein is stable in similar microsecond-length simulations at 350K and even 400K, and that there is no substantial difference between wild type and C152W at least on the time scales we could probe here (Fig 3B).

The results suggest that ASPA C152W is not severely misfolded and appears not to be globally destabilized compared to wild-type ASPA. Therefore, we speculated whether the increased proteasomal degradation of ASPA C152W might be caused by extended exposure of a degron that in the native fold is buried within the hydrophobic core of ASPA. We used the Limbo algorithm [52] to examine predicted binding sites of DnaK, the *Escherichia coli* Hsp70 homolog, and found that there were strong chaperone binding sites near position C152 (Fig 3C). To examine if this region indeed contains a potential degron, the cDNA encoding 28 amino acid residues surrounding C152 (N-RNNFLIQMFHYIKTSLAPLP<u>C</u>YVVYLIEHP-C) with or without the C152W variant was expressed in-frame with Ura3-HA-GFP. This sequence, which is mostly buried inside the folded protein (Fig 3A), appeared to be a strong degron regardless of the C152W substitution as the growth of yeast cells expressing the degron constructs was minimal without uracil (S3A Fig) and the protein levels were reduced (S3B Fig).

Prolonged exposure during *de novo* folding or transient display in the native state of one or more degrons might lead to increased recognition of C152W by components of the PQC system. Therefore, we next examined the components required for proteasomal degradation of ASPA C152W.

## ASPA degradation depends on Hsp70 and Hsp110 chaperones and the E3 ubiquitin-protein ligase Ubr1

The main substrate specificity in the process of proteasomal degradation is provided by E3 ubiquitin-protein ligases, which in the case of misfolded proteins often cooperate with the

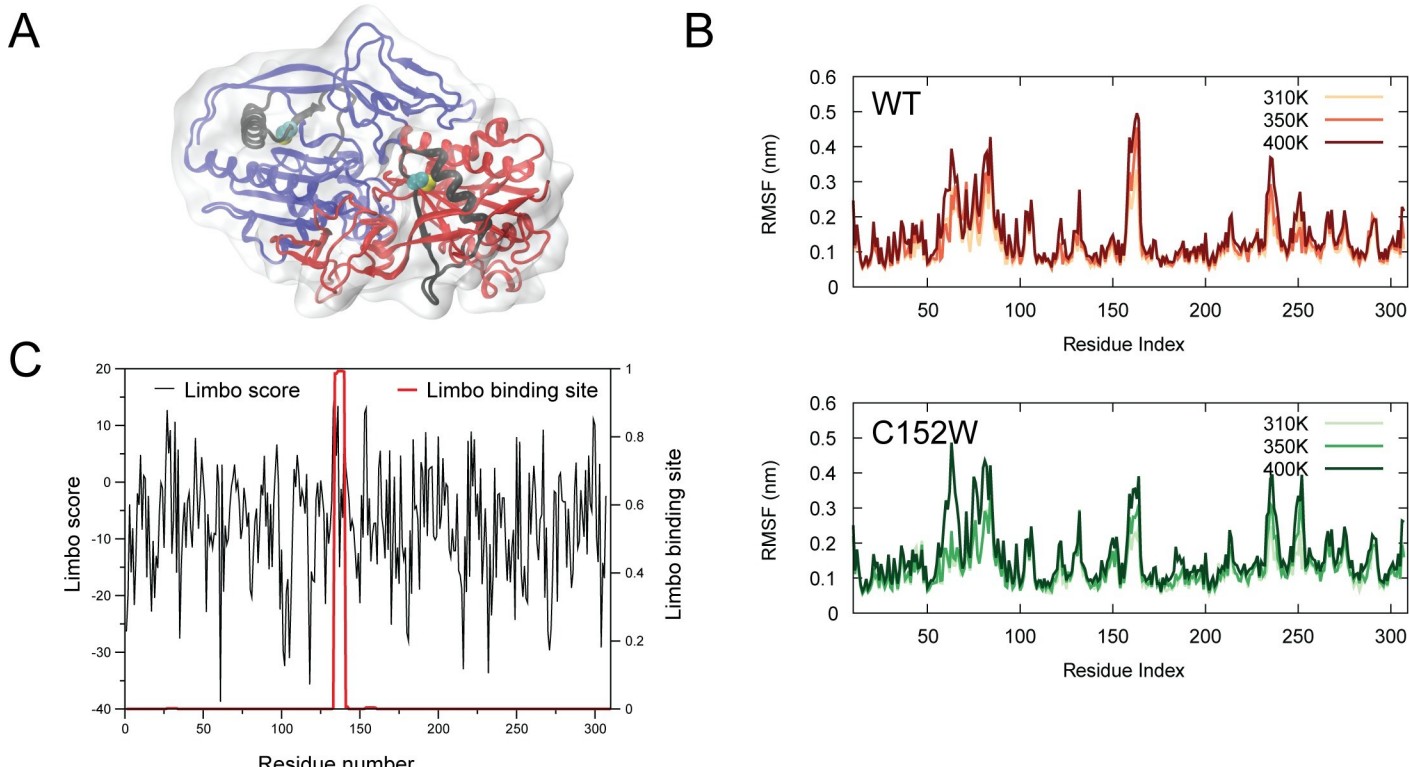

**Fig 3. ASPA C152W is not globally destabilized and might instead transiently expose a buried degron.** (A) Structure of the ASPA dimer (PDB ID 2O53) with monomers shown in red and blue. Residues 133–160, that include a putative degron, are coloured in black, and the $C_\alpha$ and heavy atoms in the side chain of C152 are shown as spheres. The protein structure is visualized with PyMOL [50] and VMD [51]. (B) We calculated the root mean square fluctuations (RMSF) of the $C_\alpha$ atoms during MD simulations of (top) wild-type and (bottom) C152W ASPA at three temperatures. The RMSF values were averaged over the results of the two monomers. (C) We used Limbo [52] to predict putative DnaK (Hsp70) binding sites in the ASPA sequence, and show both (black) the raw binding scores (over seven residue peptides) and (red) a "Boltzmann" probability to highlight the most likely binding sites.

molecular chaperone Hsp70 for substrate recognition [42]. Accordingly, we examined the involvement of the two main yeast Hsp70 chaperones, Ssa1 and Ssa2, in mediating ASPA degradation (Fig 4A) [42]. In the absence of either Ssa1 or Ssa2, cells expressing ASPA C152W grow faster on medium lacking uracil, suggesting that these Hsp70s promote degradation of ASPA C152W (Fig 4B). Accordingly, increased levels of ASPA C152W were observed in the *ssa1Δ* and *ssa2Δ* strains (Fig 4C). The phenotypes of single deletion of *SSA1* and *SSA2* are typically mild due to redundancy and compensatory expression of other Hsp70s in yeast [53]. Thus, Hsp70 appears to play a substantial role in the degradation of ASPA C152W.

In budding yeast the proteasomal degradation of Hsp70-substrates depends on the Hsp70 nucleotide exchange factor (NEF) Hsp110, encoded by the *SSE1* and *SSE2* genes [54–57]. Therefore, we next examined the participation of Hsp110 in ASPA degradation. We used a temperature-sensitive double-mutant strain, *sse1-200 sse2Δ* in which the *sse1-200* allele is functional at 25˚C, but becomes non-functional at 30˚C [58]. Absence of Hsp110 function in the *sse1-200 sse2Δ* strain at 30˚C resulted in prominent accumulation of ASPA C152W (Fig 4D), thereby substantiating the importance of the Hsp70 system in ASPA degradation. The ASPA C152W protein that accumulated in absence of Hsp110 function was insoluble (Fig 4E). There was no difference between the degradation of ASPA C152W at the two temperatures in absence of translation, indicating that Hsp110 and thus Hsp70 are mainly required for the degradation of newly synthesized ASPA (S4 Fig). This potentially suggests that the Hsp70 system

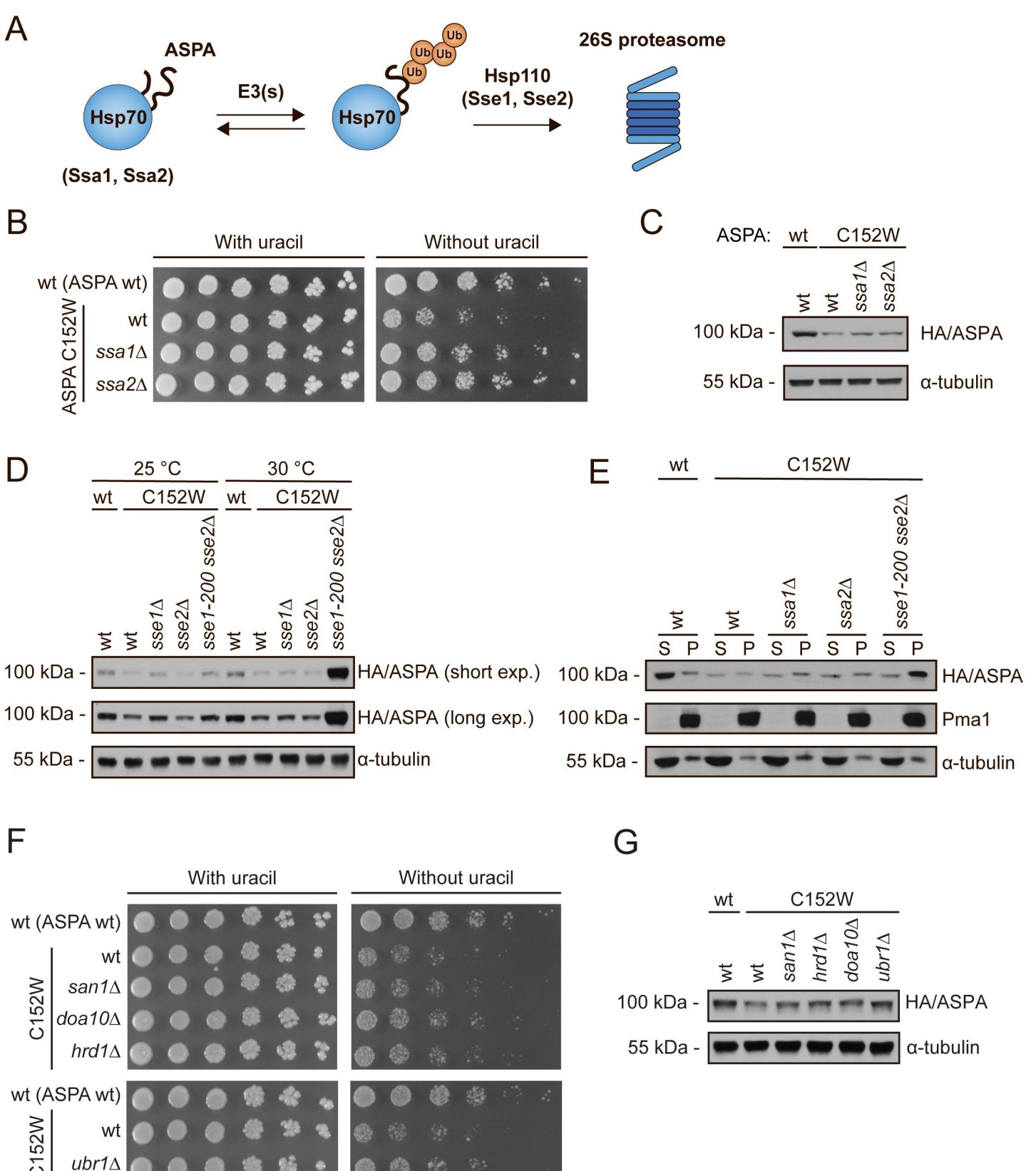

**Fig 4. Proteasomal degradation of ASPA C152W is mediated by Hsp70, Hsp110, and the E3 ubiquitin-protein ligase Ubr1.** (A) Illustration of Hsp70-mediated ASPA degradation in budding yeast. Hsp70 recognizes unfolded and misfolded proteins and interacts with various E3 ubiquitin-protein ligases, potentially resulting in substrate ubiquitination. Subsequently, Hsp70 escorts the ubiquitinated protein to the proteasome where the nucleotide exchange factor (NEF) Hsp110 promotes substrate release, thereby resulting in proteasomal degradation of the substrate protein. (B) Growth of serial-diluted yeast cells of the indicated strains expressing either wild-type ASPA or the C152W variant on solid medium with or without uracil. (C) Western blot showing ASPA protein levels in the indicated strains expressing either wild-type ASPA or C152W in presence of 0.1 mM CuSO$_4$. Blotting for α-tubulin was used as a loading control. (D) The indicated yeast strains expressing wild-type ASPA or the C152W variant were grown either at 25˚C or 30˚C prior to protein extraction and Western blotting to examine ASPA protein levels. Tubulin serves as a loading control. (E) The shown strains expressing either wild-type ASPA or the C152W variant were grown at 30˚C. Soluble (S) and insoluble (P) protein fractions were separated by centrifugation prior to Western blotting. Pma1 and α-tubulin served as loading controls for the insoluble and soluble fractions, respectively. (F) The indicated yeast strains expressing either wild-type ASPA or C152W were serial-diluted and grown on solid medium with or without uracil. (G) Western blot of ASPA protein levels in the shown strains expressing either wild-type ASPA or C152W. Tubulin is used as a loading control.

mainly targets an unstable ASPA C152W folding intermediate during *de novo* folding, rather than a structurally perturbed folded state.

To identify E3 ubiquitin-protein ligases involved in the degradation of ASPA C152W, we examined the ASPA protein levels in strains lacking the most prominent PQC E3s. The most pronounced effect was observed when ASPA C152W was expressed in a *ubr1Δ* strain (Fig 4F and 4G). In agreement with the present understanding that multiple PQC E3s display overlapping substrate specificities [40,41] stabilization was also observed in *doa10Δ* and *hrd1Δ* strains (Fig 4F and 4G). Thus, ASPA C152W displays an E3 ubiquitin ligase profile that is consistent with being a PQC substrate.

## The yeast disaggregase Hsp104 promotes solubilisation of ASPA C152W

Since ASPA formed aggregates, we hypothesized that another molecular chaperone potentially affecting ASPA steady-state levels could be the yeast disaggregase Hsp104 [59]. In addition to their roles in protein folding and degradation, Hsp70 and Hsp110 recruit Hsp104 to aggregated proteins [58,60], and promote disaggregation of insoluble protein inclusions, thus allowing aggregated proteins to attempt refolding (Fig 5A) [61]. Expression of ASPA C152W in *hsp104Δ* cells resulted in strongly reduced growth on medium lacking uracil and consistently, reduced protein levels of both ASPA variants (Fig 5B and 5C). Importantly, when we examined the solubility of ASPA in the *hsp104Δ* strain, we found that absence of Hsp104 resulted in a reduction of soluble ASPA C152W (Fig 5D). Moreover, we observed a general decrease in the level of also insoluble ASPA protein in absence of Hsp104 (Fig 5D). These results show that Hsp104 plays a key role in maintaining ASPA protein levels and suggest that ASPA as well as the C152W variant are substrates of Hsp104.

Since also the levels of aggregated ASPA decreased in *hsp104Δ* cells it suggested that another parallel disaggregase system was active on the aggregated species. In addition to Hsp104, yeast cells utilize the segregase Cdc48 to disaggregate ubiquitinated protein aggregates for proteasomal degradation (Fig 5A) [62,63]. We tested if Cdc48 was involved by expressing both ASPA variants in *cdc48-1* cells. Indeed, both ASPA variants were stabilized in the *cdc48-1* cells (S5A Fig). Specifically, the levels of insoluble ASPA protein were increased (S5B Fig). Although reduced Cdc48 function did not appear to affect the appearance of ASPA C152W aggregates (S5C Fig), there were significantly more cells containing aggregates when ASPA C152W was expressed in the *cdc48-1* background compared to a wild-type strain (S5D Fig). The number of aggregates within these cells was similar (S5E Fig). Collectively, these results imply that Cdc48 promotes the degradation of insoluble ASPA protein, and when Cdc48 function is reduced, aggregates are not as efficiently cleared.

We next examined how Hsp104 affected ASPA aggregation using fluorescence microscopy. Notably, the lack of Hsp104 lead to an accumulation of cells containing multiple aggregates (Fig 5E and 5G). In contrast, Hsp104 had no significant effect on the number of cells

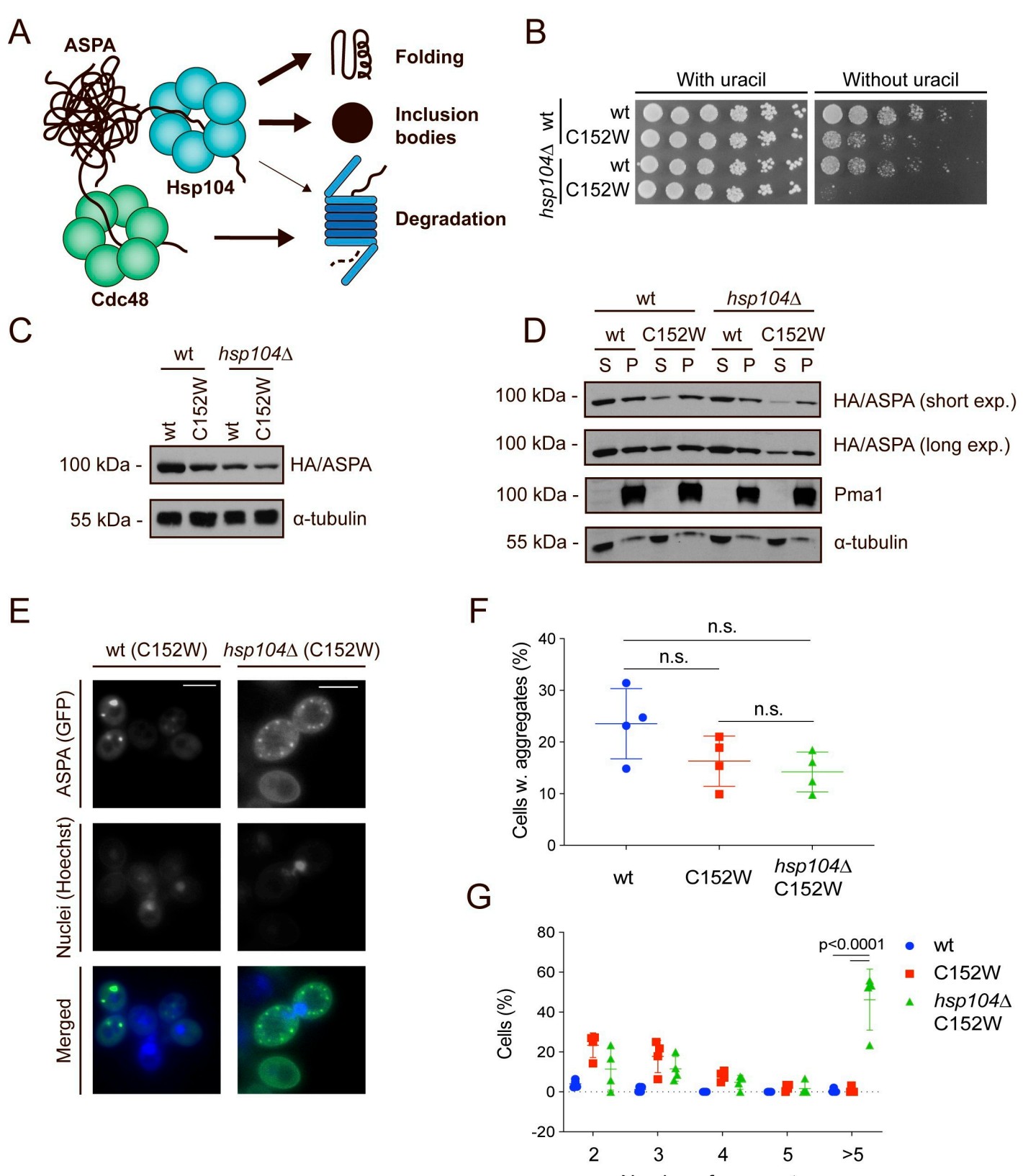

**Fig 5. Hsp104 dissolves ASPA aggregates to promote protein folding and inclusion body formation.** (A) Illustration showing potential outcomes of Hsp104-mediated disaggregation. Hsp104 dissolves protein aggregates, thereby promoting protein folding, sorting of aggregates into inclusion structures, and proteasomal degradation. In addition, the segregase Cdc48 facilitates proteasomal degradation of protein aggregates. (B) Growth of serial-diluted yeast cells of the indicated strains expressing either wild-type ASPA or the C152W variant on medium with or without uracil. (C) Western blot of ASPA protein levels in the indicated strains expressing either wild-type ASPA or C152W in presence of 0.1 mM CuSO$_4$. Blotting for α-tubulin was used as a loading control. (D) The indicated ASPA constructs were expressed in wild-type or *hsp104Δ* yeast cells in presence of 0.1 mM CuSO$_4$ and soluble (S) and insoluble (P) protein was separated by centrifugation prior to Western blotting. Pma1 and α-tubulin serve as loading controls for the insoluble and soluble fractions, respectively. (E) Representative fluorescence microscopy images of the indicated strains expressing ASPA C152W. Hoechst dye was used to stain cell nuclei. Scale bars are 4 μm. (F) Cells expressing GFP and containing aggregates were quantified in wild-type (wt, C152W) and *hsp104Δ* yeast cells expressing the indicated ASPA construct using fluorescence microscopy. The frequencies of cells containing aggregates in separate experiments are shown as data points (minimum of 153 GFP-expressing cells, n = 4). Aggregate frequencies were compared by a one-way ANOVA and Tukey's multiple comparisons test using GraphPad Prism. Means and standard deviations are shown. (G) Cells containing aggregates and the number of aggregates were quantified in wild-type (wt, C152W) and *hsp104Δ* yeast cells expressing the shown ASPA construct using fluorescence microscopy. The frequencies of each observation in independent experiments are shown as data points (minimum of 153 GFP-expressing cells, n = 4). GraphPad Prism was used to compare the number of aggregates using a two-way ANOVA and Tukey's multiple comparisons test. Means and standard deviations are indicated.

containing aggregates (Fig 5F). This is consistent with a previous study showing that proteins initially form small amorphous stress foci, and then depend on Hsp104 to remodel these smaller aggregates into larger inclusions [64]. In addition, a recent study in fission yeast found that chaperone-mediated sequestration of misfolded proteins into protein aggregate centres (PACs) protected proteins from degradation and allowed for subsequent refolding [65]. Hence, without Hsp104, ASPA C152W fails to undergo refolding-linked disaggregation and is routed by parallel pathways to degradation or remains in the smaller inclusions.

## The degradation pathway is conserved in human cells

Finally, we wanted to examine whether the degradation pathway identified in yeast cells is relevant also in human cells. Therefore, we next probed the identified degradation components in the human U2OS cell line that we transiently transfected with *ASPA* cDNA fused 5' to an RGS·His tag. First, we tested whether the ASPA C152W variant also showed a reduced protein level in human cells. As expected, the steady-state level of ASPA C152W was reduced compared to wild-type ASPA when examined by Western blotting (Fig 6A). The decreased steady state level of C152W was a result of an increased turnover since the protein level was dramatically reduced when translation was blocked by cycloheximide (Fig 6B). Furthermore, ASPA degradation was ubiquitin-dependent (S6 Fig) and was, as in yeast, mediated by the proteasome (Fig 6C). Having established that ASPA C152W is also a proteasome target in human cells, we continued to examine the role of Hsp70 in ASPA degradation by treating cells with the Hsp70 inhibitor YM-01. Inhibiting Hsp70 resulted in strong stabilization of ASPA C152W (Fig 6D), showing that Hsp70 also mediates ASPA degradation in human cells. To further corroborate the role of Hsp70, we examined whether the Hsp70 NEF Hsp110 affected ASPA steady-state levels. There are three cytosolic Hsp110 homologs in human cells, and we targeted two of these (*HSPH1* and *HSPA4*) individually using siRNA, which resulted in accumulation of both ASPA variants (Figs 6E and S7A). The degradation of ASPA C152W could be restored through complementation with a plasmid encoding siRNA-resistant *HSPH1* (S8A Fig). Knock-down of *HSPH1* did not result in detectable changes in the levels of Hsp70 or ubiquitin-protein conjugates (S8B Fig). Collectively, these results suggest that Hsp110 is important for degradation of ASPA. When we examined the solubility of the two ASPA variants before and after *HSPH1* and *HSPA4* knock-down, we found that the disease-linked variant C152W displayed a similar behaviour as in yeast cells. It was completely insoluble, and accordingly also the accumulated protein in *HSPH1* and *HSPA4* knock-down cells was found in the insoluble fraction (Figs 6F and S7B).

In budding yeast, Hsp110 mediates degradation of Hsp70 substrates by interacting with the 19S regulatory particle of the 26S proteasome where it accelerates the release of Hsp70

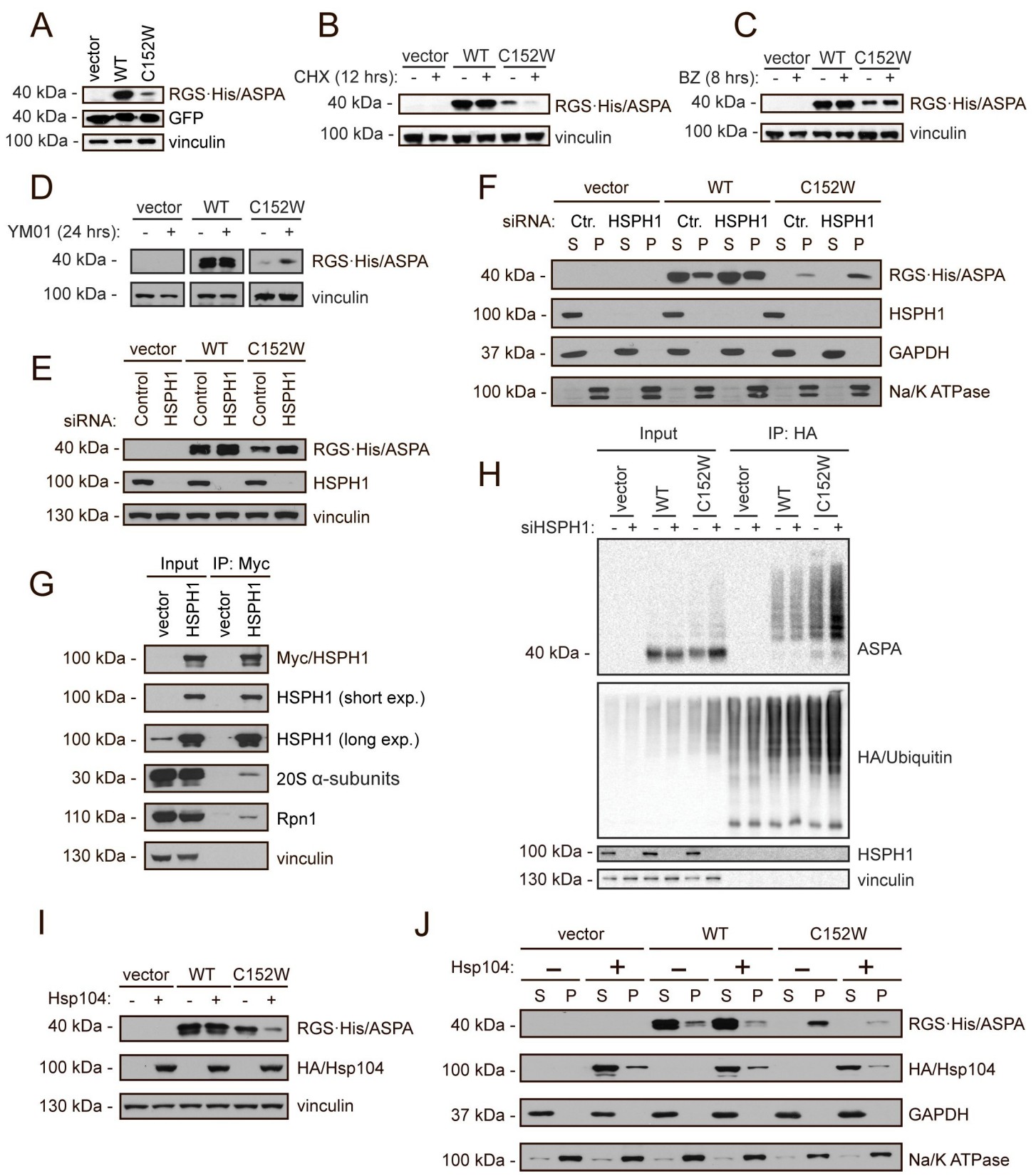

**Fig 6. Chaperone-mediated proteasomal degradation of ASPA C152W in human cells.** (A) U2OS cells were transiently transfected with the indicated constructs and a GFP expression plasmid to normalize for differences in transfection efficiency. ASPA and GFP protein levels were visualized by Western blotting. Vinculin was used as a loading control. (B) Western blot showing ASPA protein levels in U2OS cells transiently transfected with the indicated constructs prior to treatment with the translation inhibitor cycloheximide (CHX) for 12 hours (+) or the solvent DMSO as a control (-). Vinculin serves as a loading control. (C) U2OS cells were transiently transfected with the indicated constructs prior to treatment with the proteasome inhibitor bortezomib (BZ) for 8 hours (+) or, as a control, the solvent DMSO (-). ASPA protein levels were examined by Western blotting. Blotting for vinculin is used as a loading control. (D) Transiently transfected U2OS cells were treated with the Hsp70 inhibitor YM-01 for 24 hours (+) or the solvent DMSO (-) prior to protein extraction and Western blotting. Vinculin is used as a loading control. (E) U2OS cells were reverse transfected with the shown siRNAs, followed by transient transfection with the constructs shown at the top. Then, ASPA protein levels were examined by Western blotting. Vinculin serves as a loading control. (F) The soluble (S) and insoluble (P) fractions of protein were separated in cell extracts treated as in E prior to Western blotting. GAPDH and the Na/K ATPase serve as a loading control for the soluble and insoluble protein fraction, respectively. (G) U2OS cells were transiently transfected with HSPH1-Myc prior to Myc immunoprecipitation followed by Western blotting. (H) U2OS cells were reverse transfected with control siRNA (-) or siRNA targeting *HSPH1* (+) prior to transfection of HA-ubiquitin and wild-type ASPA or C152W as indicated. The cells were used for denaturing HA immunoprecipitation followed by Western blotting. (I) U2OS cells were transiently co-transfected with Hsp104 and the indicated constructs, and ASPA protein levels were examined by Western blotting. Blotting for vinculin is used as a loading control. (J) Cells were treated as in I followed by differential centrifugation to separate soluble (S) and insoluble (P) protein fractions and Western blotting. Blotting for GAPDH and the Na/K ATPase is used as a loading control for the soluble and the insoluble fractions respectively.

substrates and thus promotes their degradation [54]. However, a similar role for Hsp110 in human cells has not been examined. We hypothesized that Hsp110 might mediate degradation of ASPA in human cells through a similar mechanism, and therefore examined whether HSPH1 interacted with the 26S proteasome in human cells. Myc-tagged HSPH1 was expressed in U2OS cells, and HSPH1 was isolated using immunoprecipitation, followed by Western blotting, to examine a potential interaction with the proteasome. Both 20S proteasome core α-subunits and the 19S regulatory subunit Rpn1 co-immunoprecipitated with HSPH1 (Fig 6G). Hence, similar to the situation in yeast cells, Hsp110 also interacts with the 26S proteasome in human cells, presumably to catalyse release of Hsp70-bound clients for degradation [54]. To further substantiate that Hsp110 mediates ASPA degradation at the proteasome, we examined at which step Hsp110 acts during ASPA C152W degradation. We co-expressed ASPA and HA-tagged ubiquitin in U2OS cells treated with siRNA targeting *HSPH1*, and isolated ubiquitin and ubiquitin-protein conjugates by immunoprecipitation. To ensure that only ubiquitin-conjugated proteins and not ubiquitin-interacting proteins were isolated, the lysates were denatured prior to precipitation. Following knock-down of *HSPH1*, we observed an increase in ubiquitinated ASPA C152W (Fig 6H), suggesting that HSPH1 promotes ASPA degradation following ubiquitination, consistent with Hsp110-stimulated release of ASPA from Hsp70 at the proteasome.

In yeast, Hsp104 appeared to mediate refolding of aggregated ASPA C152W, thereby allowing more soluble ASPA protein to form. Since there is no known metazoan homolog of Hsp104, we attempted to express yeast Hsp104 in human cells to promote solubilisation of ASPA C152W. We found that expression of Hsp104 resulted in a decreased level of ASPA C152W (Fig 6I). When we examined the solubility of ASPA, we found that Hsp104 specifically decreased the level of insoluble ASPA protein for both variants (Fig 6J). Yeast Hsp104 therefore appears to still function as a disaggregase when expressed in human cells, and might promote ASPA degradation by dissolving aggregated protein [66–68].

## Discussion

Based on our studies in yeast, we propose a model for the degradation of ASPA C152W illustrated in Fig 7. The C152W substitution causes increased proteasomal degradation of ASPA facilitated by increased recognition by the PQC system. Although the C152W variant does not appear to perturb the ASPA native structure substantially, we cannot rule out that the C152W substitution does not lead to a transient exposure of a buried degron. An alternative, and perhaps more likely, explanation is that the variant displays a delayed *de novo* folding, thus trapping the protein in an unstable folding intermediate during translation. The fraction of the

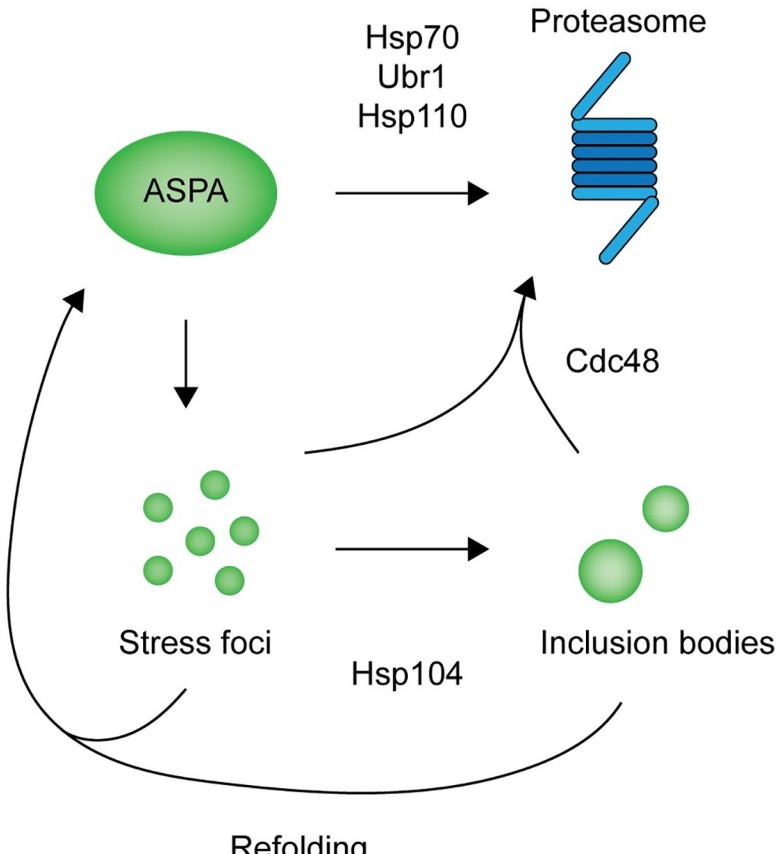

**Fig 7. Illustration of our model for ASPA degradation in budding yeast.** ASPA C152W is targeted to the proteasome for degradation by Hsp70, Hsp110, and E3 ubiquitin-protein ligases including Ubr1. In addition, ASPA C152W forms aggregates that are dissolved by Hsp104 to promote sorting of ASPA into inclusion structures as well as refolding to form soluble ASPA. Finally, Cdc48 mediates degradation of insoluble ASPA protein.

C152W variant that makes it to the native conformation is thus stable, while the bulk of the synthesized proteins is degraded. Regardless of the exact mechanism, in both cases the degron would be available for recognition by the PQC system, presumably involving Hsp70. This in turn, results in increased degradation by the 26S proteasome and consequently a decreased steady-state level of ASPA C152W compared to wild-type ASPA. Proteasomal degradation of ASPA C152W is mediated by Hsp70, the Hsp70 NEF Hsp110, and the E3 ligase Ubr1, likely along with other E3s. Both wild type and the C152W ASPA variant form insoluble aggregates, although C152W shows a higher aggregation propensity. The aggregates appear to be disaggregated by Hsp104 to increase protein folding and sorting into larger inclusions. Finally, we observed that Cdc48 promotes degradation of insoluble ASPA protein. When we examined ASPA degradation in mammalian cells, the increased proteasomal turnover of ASPA C152W was similarly mediated by the Hsp70 system, including Hsp110.

A previous study found that the C152W variant has thermal and conformational stability comparable to wild-type ASPA but shows reduced specific activity [21]. In contrast, we find a reduced steady-state of ASPA C152W *in vivo*. This discrepancy potentially stems from the differences in studying protein stability *in vitro* and *in vivo*. While the C152W variant might not cause global destabilization of the ASPA structure in agreement with *in vitro* measurements, local unfolding or defects in protein folding during translation might trigger degradation of

ASPA mediated by the cellular PQC system and consequently reduced stability *in vivo*. It has also not been systematically established how loss of thermal stability and recognition by the PQC system are linked. As the PQC system does not directly measure thermal stability, there may be cases where degradation is triggered despite similar overall stability.

In this study, we did not examine any mammalian E3 ubiquitin-protein ligases that might mediate ASPA degradation. If, however, an E3 important for ASPA degradation could be identified, it could potentially be a promising therapeutic target for Canavan disease. In yeast, we found that the E3 Ubr1, to some extent, promoted ASPA degradation. In addition, Ubr1 cooperates with Hsp70 for substrate ubiquitination [56], consistent with our observation that ASPA degradation is Hsp70-dependent. Although there are Ubr1 orthologues in humans, some of which are implicated in the N-end rule pathway [69,70], like the yeast orthologue [71], human UBR E3 ligases have not, to our knowledge, been connected to the cytosolic PQC system as in yeast [56,72,73]. These human Ubr1 orthologues are likely to display overlapping substrate specificity, perhaps also with other PQC E3s, including CHIP (also known as STUB1), which interacts directly with molecular chaperones [74] and functions as an E3 ubiquitin ligase in cytosolic PQC [75,76]. Based on our results, showing that Hsp70 and Hsp110 mediate degradation of ASPA C152W, the human Ubr1 orthologues and CHIP may all, to some extent, be involved in ASPA degradation in human cells.

In addition, it remains to be examined whether the degradation pathway identified for ASPA C152W is also relevant for other destabilized ASPA variants. To this end, we propose that the multiplexed genetic assay variant abundance by massively parallel sequencing (VAMP-seq) could be applied [77]. In this way, the steady-state level of every ASPA variant could be examined, and the effect of targeting e.g. Hsp70 and Hsp110 could be identified, to uncover the universality of the degradation pathway mapped using the C152W variant. In addition, such protein abundance data could be combined with an assay testing ASPA function to identify the variants for which it is relevant to target the PQC system to prevent loss of function. Although it is likely relevant for some ASPA variants to target the PQC system, this might not be the case for the C152W variant. Previous studies have been unable to detect any activity for ASPA C152W when expressed in COS-7 cells [15,17], while very low activity could be detected when expressed in *Pichia pastoris* [21].

Since we observed a stabilization of both wild-type ASPA and the C152W variant upon *HSPH1* knock-down, this suggests that at least Hsp110 is likely important for the degradation of multiple ASPA variants. Based on our results, we propose that Hsp110 might promote degradation of Hsp70 substrates, including ASPA, in mammalian cells through its interaction with the proteasome. This function is similar to that of the Hsp70 NEF BAG1, which has long been known to associate with the 26S proteasome, thus providing a physical link between Hsp70 and the proteasome [78–80]. In this way, BAG1 promotes degradation of ubiquitinated Hsp70 substrates, which is similar to the function of Hsp110 in budding yeast and, as we show here, in mammalian cells [54]. Accordingly, BAG1 and Hsp110 might have redundant functions in mammalian cells. However, since we found that knock-down of either *HSPH1* or *HSPA4* was enough to stabilize ASPA C152W, such a redundancy seems unlikely at least for ASPA degradation. This suggests that BAG1 and Hsp110 may have separate substrate specificities, thus promoting degradation of different Hsp70 substrates.

In conclusion, we propose that the C152W substitution in ASPA leads to increased exposure of buried hydrophobic sequences, perhaps due to delayed *de novo* folding of ASPA, which in turn triggers extended interaction with Hsp70 leading to its targeting by various E3 ubiquitin ligases. Upon ubiquitination, ASPA C152W is escorted to the 26S proteasome, where Hsp110 promotes release from Hsp70, thereby resulting in proteasomal degradation. Due to the apparent redundancy in E3 ubiquitin-protein ligases promoting ASPA degradation, we

propose that the Hsp70 system would be a better target to limit ASPA degradation. Although targeting Hsp110 to stabilize ASPA C152W resulted in the accumulation of insoluble protein, it might still be a relevant target for other ASPA variants that are at least partially soluble and contain residual activity. Accordingly, Hsp110 should be explored further as a potential therapeutic target to decrease the degradation of ASPA and other misfolded proteins, possibly using a recently developed inhibitor of Hsp110 [81], to potentially avoid or diminish the development of Canavan disease and other protein misfolding disorders such as Lynch syndrome or Birt-Hogg-Dubé syndrome [82,83].

## Materials and methods

### Buffers

Buffer A: 25 mM Tris/HCl, 50 mM NaCl, 10% glycerol, 2 mM MgCl·6H$_2$O, pH 7.4. Prior to use, 1 mM PMSF and 1 Complete Mini Protease Inhibitor Cocktail tablet (Roche) were added to 10 mL buffer. Buffer B: 50 mM Tris/HCl, 150 mM NaCl, 1 mM EDTA, pH 7.4. Prior to use, 1 Complete Mini Protease Inhibitor Cocktail tablet (Roche) was added to 10 mL buffer. Buffer C: 25 mM Tris/HCl, 25 mM NaCl, 10% glycerol, 0.1% Triton X-100, 2 mM MgCl$_2$, pH 7.5. Prior to use, ATP was added to a final concentration of 5 mM and pH was re-adjusted to 7.5. Then, 1 Complete Mini Protease Inhibitor Cocktail tablet (Roche) was added to 10 mL buffer. Buffer D: 30 mM Tris/HCl, 100 mM NaCl, 5 mM EDTA, pH 8.1. Prior to use 0.2 mM PMSF was added. PBS: 10 mM Na$_2$HPO$_4$, 1.8 mM KH$_2$PO$_4$, 137 mM NaCl, 3 mM KCl, pH 7.4. SDS sample buffer (4x): 250 mM Tris/HCl, 8% SDS, 40% glycerol, 0.05% bromophenol blue, 0.05% pyronin G, 2% β-mercaptoethanol, pH 6.8.

### Plasmids

Full-length wild-type human *ASPA* cDNA with an N-terminal RGS6xHis-tag was expressed from pcDNA3.1 (Genscript). The C152W variant was generated by Genscript. The empty vector control was kindly provided by Dr. Mads Gyrd-Hansen. Full-length codon-optimized yeast *HSP104* fused C-terminally to an HA-tag was expressed from pcDNA3.1 (Genscript). Human codon-optimized *HSPH1* fused C-terminally to a Myc-tag was expressed from pcDNA3.1 (Genscript). cDNA encoding wild-type human ubiquitin containing an N-terminal HA-tag was expressed from pRK5-HA (Addgene). Full-length EGFP fused N-terminally to a nuclear localization signal (NLS) and C-terminally to a Myc-tag was expressed from pCMV (Clontech).

In yeast, wild-type codon-optimized *ASPA* cDNA fused to Ura3-HA-GFP at the N-terminal of *ASPA* was expressed from pTR1412 (Genscript). The pTR1412 vector was kindly provided by Dr. Tommer Ravid. The ASPA fusion constructs were used for all yeast experiments. The C152W variant was generated by Genscript. The ASPA degron sequences fused N-terminally to Ura3-HA-GFP were expressed from pTR1412, and the sequences were RNNFLIQMFHYIK TSLAPLPCYVYLIEHP (Degron wt) and RNNFLIQMFHYIKTSLALPWYVYLIEHP (Degron C152W) (Genscript).

### Cell culture

U2OS cells (ATCC) were propagated in Dulbecco's Modified Eagle Medium (DMEM) containing 10% fetal-calf serum (Invitrogen), and supplemented with 2 mM glutamine, 5000 IU/ mL penicillin, and 5 mg/mL streptomycin at 37˚C. STR analysis was employed for cell line authentication (Eurofins). FugeneHD (Promega) was used for transfections according to the manufacturer's instructions. Plasmids encoding *ASPA* were diluted 1:1 with empty vector or the plasmid to be co-expressed prior to transfection.

## Electrophoresis and blotting

SDS-PAGE was performed using 8% and 12.5% acrylamide gels. For the Western blotting procedure, 0.2 μm nitrocellulose membranes were used. Following protein transfer to membranes, these were blocked in 5% fat-free milk powder, 5 mM $NaN_3$ and 0.1% Tween-20 in PBS.

Antibodies and their sources were: anti-HA (Roche, 15645900), anti-Cdc48 (own production), anti-α-tubulin (Abcam, YL1/2 MA1-80017), anti-Pma1 (Abcam, 40B7 Ab4645), anti-RGS-His (Qiagen, 34610), anti-Vinculin (Sigma, hVIN1 V9264), anti-GAPDH (Cell signalling technology, 14C10 2118), anti-Na/K ATPase α-1 (Merck, C464.6 05–369), anti-Hsp105 (HSPH1, Abcam, Ab108625), anti-HSPA4 (Abcam, Ab185219), anti-Ubiquitin (Dako, Z0458), anti-Myc (Chromotek, 9E1 9e1-100), anti-Rpn1 (Enzo Life Sciences, p112-1 PW9270), anti-20S α-subunits (Enzo Life Sciences, MCP231 PW8195), anti-Hsp70 (Invitrogen, 5A5 MA3-007), anti-Aspartoacylase (Thermo Scientific, PA5-29180), anti-GFP (Chromotek, 3H9 3h9-100). Secondary antibodies and their sources were: HRP-anti-rat (Invitrogen, 31470), HRP-anti-mouse (Dako, P0260), HRP-anti-rabbit (Dako, P0448).

## Protein degradation experiments

Bortezomib (LC Laboratories) was used at 15 μM for 8 hours in serum-free media. YM-01 (Abcam) was used at 10 μM for 24 hours in serum-free media. Cycloheximide (Sigma) was used at 10 μg/mL for 12 hours in serum-free media.

For siRNA gene knock-downs, Lipofectamine RNAiMAX was used for reverse transfection of siRNA according to manufacturer's instructions. The following siRNAs were used: *siHSPH1* (Dharmacon, #M-004972-00-0005), *siHSPA4* (Dharmacon, #M-012636-02-0005), Non-targeting (Dharmacon, #D-001206-13-05).

## Co-precipitation of the 26S proteasome

For HSPH1 immunoprecipitation, confluent U2OS cells transiently transfected with a plasmid encoding *HSPH1* or an empty vector control were harvested in 600 μL buffer C. The cells were sonicated three times 30 seconds and were cooled on ice in between. Samples were centrifuged (12500 g, 30 min, 4˚C) to remove cell debris. Myc-Trap agarose beads (Chromotek) were washed twice in buffer C (1000 g, 1 min, RT), and 15 μL beads were transferred to a new Eppendorf tube for each sample. After centrifugation, 30 μL supernatant was added to 12.5 μL 4x sample buffer to use for input, and the remaining supernatant was transferred to the Myc-Trap beads. The samples were tumbled for two hours at 4˚C. Subsequently, samples were washed four times in 500 μL buffer C (1000 g, 1 min, 4˚C). Proteins bound to the Myc-Trap beads were eluted in 40 μL 1.5x sample buffer. Samples were boiled and examined by SDS-PAGE and Western blotting.

## Denaturing HA immunoprecipitation

For denaturing immunoprecipitation of HA-ubiquitin, confluent cells were transiently transfected with plasmids encoding HA-ubiquitin and vector or ASPA mixed 1:1. In experiments where cells were treated with bortezomib this was done before harvesting as noted above. In experiments where *HSPH1* mRNA was depleted, cells were reverse transfected with *siHSPH1* or non-targeting siRNA prior to transient transfection of plasmids. Cells were harvested in 300 μL buffer D. Samples were sonicated three times 10 seconds and were kept on ice in between. Then, 75 μL 8% SDS was added to all samples. The samples were vortexed and boiled for 10 minutes. Cooled samples were mixed with 1125 μL 2.5% Triton X-100 in buffer D, and were then left on ice for 30 minutes. The samples were centrifuged (16000 g, 60 min, 4˚C).

Meanwhile, HA agarose beads (Sigma) were washed two times in buffer D (1000 g, 1 min, RT), and 15 μL beads were transferred to a new Eppendorf tube for each sample. Following centrifugation, 30 μL supernatant was mixed with 12.5 μL 4x sample buffer for input, and the remaining supernatant was transferred to the HA agarose beads. The samples were tumbled for two hours at 4˚C. Thereafter, the beads were washed four times in buffer D containing 1% Triton X-100 and one time in buffer D (1000 g, 1 min, 4˚C). Bound proteins were eluted in 40 μL 1.5x sample buffer. Samples were boiled and analysed by SDS-PAGE and Western blotting.

## ASPA solubility

The solubility of the two ASPA variants was examined by harvesting transiently transfected U2OS cells in a 6-well plate in 200 μL buffer B. Samples were sonicated three times 20 seconds and were cooled on ice in between. Then, samples were centrifuged (15000 g, 30 min, 4˚C) and the supernatant (200 μL) was transferred to a new Eppendorf tube containing 75 μL 4x sample buffer, while the remaining pellet was resuspended in 275 μL 1.5x sample buffer. Samples were analysed by SDS-PAGE and Western blotting.

## Yeast strains

All yeast strains were obtained from the Euroscarf collection, except for the following strains. The *his3Δ1*, *leu2Δ*, *met15Δ*, *ura3Δ*, *doa10*::*kanMX* strain was kindly provided by Dr. Tommer Ravid. The *his3Δ1*, *leu2Δ*, *met15Δ*. *ura3Δ*, *hsp104-mCherry*:*kanMX* strain was kindly provided by Dr. Yves Barral. The *lys2Δ*, *suc2Δ*, *his3Δ*, *leu2Δ*, *trp1Δ*, *ura3Δ*, *mCherry-ATG8-hphNT1* strain was kindly provided by Dr. Kuninori Suzuku. The *leu2Δ*, *met15Δ*, *ura3Δ*, *sse1*::*hphMX4*, *sse2*::*kanMX4* strain was unless otherwise noted grown at 25˚C. Transformations were done using lithium acetate as described in [84]. Yeast cells were cultured in synthetic complete (SC) medium (2% D-glucose monohydrate, 0.67% yeast nitrogen base without amino acids, (0.2% drop out), (0.0225% amino acids)). Yeast minimal medium with the carbon source exchanged with *N*-acetylaspartate (NAA) consisted of 2% NAA, 0.67% yeast nitrogen base without amino acids, and 0.0225% amino acids. Yeast minimal medium with the nitrogen source replaced with NAA consisted of 2% D-glucose monohydrate, 0.17% yeast nitrogen base without amino acids and ammonium sulphate, 0.35% NAA, and 0.0225% amino acids.

## Yeast growth experiments

Growth on solid media was tested using yeast cultures incubated overnight (30˚C, vigorous agitation) until reaching stationary phase. Yeast cells were washed in sterile water (1200 g, 5 min, RT) and resuspended in sterile PBS. All cultures were adjusted to an $OD_{600nm}$ of 0.40 and were then used for a five-fold serial dilution in PBS, which was applied in 5 μL drops on agar plates. The plates were briefly dried and incubated at 30˚C, unless otherwise noted, for two to four days.

Growth in liquid media was examined using cultures incubated overnight (30˚C, vigorous agitation) until reaching exponential phase. These cultures were diluted to an $OD_{600nm}$ of 0.10 using pre-warmed 30˚C medium. The cultures were incubated for 10 hours (30˚C, vigorous agitation) and the $OD_{600nm}$ was measured every hour. Growth was compared by fitting exponential curves to the data points using the exponential growth equation in GraphPad Prism. The three replicates of each condition were used to fit one average curve as well as individual curves and doubling times were compared using a one-way ANOVA and Tukey's multiple comparisons test.

For liquid growth comparisons between the wild-type and ASPA C152W strain, pre-cultures were diluted to an $OD_{600nm}$ of exactly 0.010 and then added to a 96-well plate (Nunc) with cover in quadruplicates. Wells containing media alone, were included as a reference. The $OD_{600nm}$ was monitored in a microplate reader (Tecan Infinite 200 PRO) at 30°C for 24 hours.

### Protein extraction from yeast cells

Proteins were extracted for Western blotting using trichloroacetic acid (Sigma) and glass beads as described in [85].

ASPA solubility was examined by harvesting yeast cells in exponential phase (1200 g, 5 min, 4°C). The cell pellets were washed in water (1200 g, 5 min, 4°C), and were subsequently resuspended in 200 μL buffer A and transferred to 2 mL screw cap tubes containing 0.5 mL glass beads (Sigma Life Science). The cells were lysed using a Mini Bead Beater (BioSpec Products) by three 10 second cycles with 10 minute incubations on ice between each burst. The tubes were punctured at the bottom using a needle and were placed in 15 mL falcon tubes containing a 1.5 mL Eppendorf tube without the lid. Samples were transferred to the Eppendorf tubes by centrifugation (180 g, 1 min, 4°C), which were then centrifuged (2000 g, 2 min, 4°C). Subsequently, 150 μL of the supernatant was transferred to an Eppendorf tube and samples were centrifuged (13000 g, 30 min, 4°C) to separate soluble and insoluble protein. The supernatant was mixed with 50 μL 4x sample buffer, while the pellet was resuspended in 200 μL 1.5x sample buffer. The samples were boiled for 5 minutes and examined using SDS-PAGE and Western blotting.

Bortezomib (BZ) (LC Laboratories) was used at a final concentration of 1 mM. For growth assays on solid media, bortezomib was added to melted and cooled agar. For liquid cultures, bortezomib was added to cultures in exponential phase which were then incubated for three hours (30°C, vigorous agitation). Cycloheximide (CHX) (Sigma) was used at a final concentration of 100 μg/mL. Cycloheximide dissolved in DMSO was added to cultures in exponential phase which were incubated for the time indicated in figures.

### Microscopy

Microscopy images of yeast cells were acquired using Volocity Software (PerkinElmer) and a widefield microscope (Axio-Imager Z1; Carl Zeiss) equipped with a 100x objective lens (Carl Zeiss), differential interference contrast (DIC), a cooled charge-coupled device (CCD) camera (Hamamatsu Photonics), and an illumination source (Carl Zeiss). Cultures in exponential phase induced with 0.1 mM $CuSO_4$ were harvested by centrifugation (1200 g, 5 min, RT). The resulting cell pellet was resuspended in a small volume of supernatant, and 5 μL cell suspension was transferred to a microscopy slide and was used for fluorescence microscopy. The fluorescence microscopy images were processed using ImageJ [86]. Cell nuclei were stained using Hoechst stain (Sigma-Aldrich), which was added to cells in exponential phase at a final concentration of 5 μg/mL, followed by 10 minutes of incubation with shaking. Cells were then harvested and washed in fresh medium (900 g, 5 min, RT), and prepared for microscopy as above. Protein aggregates were counted manually, and statistics and graphs were made using Graph-Pad Prism. The percentage of GFP signal found in aggregates was quantified using ImageJ. Only cells containing aggregates were used for the quantification. The total GFP signal in each cell was calculated by multiplying the area of each cell with the mean grey (GFP) intensity of the cell. The GFP signal found in each aggregate was calculated by multiplying the aggregate area with the mean grey (GFP) intensity of the aggregate. If multiple aggregates were found in a single cell, the GFP signal of each aggregate was summed. Then for each cell the GFP signal in aggregates was divided by the total GFP signal, and a percentage was calculated. The

percentage of GFP signal found in aggregates was compared for cells expressing wild-type ASPA or C152W using a Welch's t test in GraphPad Prism.

## Molecular dynamics simulations

We used a crystal structure of ASPA (PDB ID 2O53 [20]) as starting point for our MD simulations. All simulations where performed using a GPU-accelerated version of Gromacs 2018.6 [87] and used the Amber ff99SB-*disp* force field [88]. The protein dimer was solvated in a cubic box of 45300 water molecules in 150mM NaCl after removing the phosphate atoms bound to the protein in the crystal. We used the v-rescale thermostat [89] with a 1 ps coupling constant to keep the temperature constant (at different temperatures as specified above), and simulated the system in the canonical ensemble. The PME algorithm was used for electrostatic interactions. A single cut-off of 1.0 nm was used for both the PME algorithm and Van der Waals interactions. Protein structures were visualized with PyMOL [50] and VMD [51]. RMSD and RMSF of the trajectories were analysed using Gromacs gmx tools.

## Protein purification from *E. coli* and ASPA assays

*E. coli* BL21(DE3) (New England Biolabs) transformed with a pET11a-based vector for expression of 6His-tagged aspartase AspA (Genscript) was inoculated in 200 mL LB media supplemented with ampicillin at 37˚C. When $OD_{600nm}$ reached 0.5, IPTG was added to a final concentration of 1 mM and the culture was incubated for a further 3 hours before harvesting the cells. The cell pellet was resuspended in 35 mL Lysis buffer (50 mM $NaH_2PO_4$, 300 mM NaCl, pH 7.4, 1 mM PMSF and Complete protease inhibitors without EDTA (Roche)) and sonicated on ice. After centrifugation, the supernatant was passed over a TALON resin (Clontech) and purified according to the manufacturer's instructions. Fractions containing the AspA-6His protein were pooled and subsequently dialyzed against Assay Buffer (50 mM Hepes pH 7, 100 mM NaCl). By SDS-PAGE analysis AspA was estimated to be >95% pure. The protein concentration was determined by a bicinchoninic acid (BCA) assay (Pierce) using bovine serum albumin (BSA) as standard.

The ASPA wild-type and ASPA C152W were produced as GST fusion proteins in *E. coli* BL21(DE3) from the pDEST15 vector (Invitrogen) carrying the *ASPA* and *ASPA C152W* genes (Genscript) codon optimized for expression in *E. coli*. The transformed cells were cultured in 200 mL LB media supplemented with ampicillin at 25˚C. When $OD_{600nm}$ reached 0.3, IPTG was added to a final concentration of 1 mM and the culture was incubated for a further 5–6 hours before harvesting the cells. The cell pellet was resuspended in 10 mL binding buffer (25 mM Tris/HCl pH 7.4, 50 mM NaCl, 1 mM DTT, 10% glycerol, 0.5% Triton X-100, and Complete protease inhibitors without EDTA (Roche)) and sonicated on ice. After centrifugation, the supernatant was dialyzed to Assay Buffer (50 mM Hepes pH 7, 100 mM NaCl, 1 mM DTT) overnight at 4˚C.

A coupled assay devised for the characterization of human brain ASPA was used [7]. Here the production of aspartate by ASPA is coupled to the deamination and release of fumarate by *E. coli* aspartase (AspA) monitored at 240 nm ($\varepsilon_{240nm}$ = 2.53 mM$^{-1}$ cm$^{-1}$) using a Zeiss Specord S10 diode array spectrophotometer. The assay conditions were 50 mM Hepes, pH 7.4, 100 mM NaCl, 1 mM DTT. To a quartz cuvette at 25˚C, was added buffer as above, aspartase (9 μg) and *N*-Acetyl-L-aspartate (Sigma) or *N*-Chloroacetyl-L-aspartate (Sigma) to 3 mM. The reaction was initiated by addition of extracts of *E. coli* (described above). The reactions were followed over time up to 20 minutes for detection of any significant increase in absorbance at 240 nm above background levels.

## Supporting information

**S1 Fig. Characterization of ASPA aggregates.** (A) The percentage of total GFP signal found in aggregates in individual cells expressing either wild-type ASPA or C152W was quantified in 100 cells. Each data point represents the signal distribution in a single cell. The GFP signal found in aggregates was compared between wild-type ASPA and C152W using Welch's t test. Means and standard deviations are shown. (B) Wild-type yeast cells expressing either wild-type ASPA or the C152W variant grown on solid medium containing uracil and without or with copper as indicated. (C) Wild-type yeast expressing the indicated constructs were grown in liquid medium containing uracil without or with copper as indicated, and the $OD_{600nm}$ was measured at the indicated time points. Growth was compared using Tukey's multiple comparisons test on three independent cultures for each condition. There was no significant difference between the doubling time of wild-type ASPA or the C152W variant in absence nor in presence of copper. The doubling times in absence and presence of copper were significantly different (adjusted p-value < 0.0001, n = 3) possibly caused by copper itself or the high expression level of an exogenous protein. (D) Representative fluorescence microscopy images of wild-type yeast expressing GFP-ASPA C152W and mCherry-Atg8. Only rarely did we observe co-localization between ASPA aggregates and Atg8, indicating that aggregated ASPA C152W does not accumulate in the IPOD [49]. Scale bar is 4 μm.
(TIF)

**S2 Fig. AspA and ASPA produced in E. coli, and attempt at a yeast-based ASPA assay.** (A) As a helper enzyme to convert produced aspartate into fumerate for an ASPA enzyme assay [7], 6His-tagged *E. coli* AspA was produced in *E. coli* and purified. A sample of the purified material was resolved by SDS-PAGE and the gel was stained with Coomassie Brilliant Blue (CBB). (B) ASPA and ASPA C152W was produced in *E. coli* fused N-terminally to GST. The material was resolved by SDS-PAGE and the gel was stained with Coomassie (CBB). (C) Wild type yeast cells transformed with a control plasmid (vector) or an expression plasmid for wild-type ASPA were streaked as indicated on minimal media plates where either the carbon source (upper panel) or the nitrogen source (lower panel) was exchanged with *N*-acetylaspartate (NAA).
(TIF)

**S3 Fig. The region surrounding position C152 is a degron.** (A) Growth of wild-type yeast cells expressing an empty vector control or potential degrons consisting of 28 amino acids surrounding the C152 residue in the ASPA sequence without (Degron wt) or with (Degron C152W) the C152W substitution. (B) Western blot showing the protein levels of the degron constructs. Tubulin served as a loading control.
(TIF)

**S4 Fig. Hsp110 is primarily required for degradation of newly synthesized ASPA.** The *sse1-200 sse2Δ* strain expressing ASPA C152W was incubated at 25˚C until exponential phase. The culture was then split and cells were incubated at the permissive temperature (25˚C) or restrictive temperature (30˚C) with the translation inhibitor cycloheximide (CHX) or as a control the solvent DMSO. Cells were harvested at the indicated time points, followed by protein extraction and Western blotting to examine the steady state of ASPA C152W. Tubulin served as a loading control.
(TIF)

**S5 Fig. Cdc48 promotes degradation of insoluble ASPA.** (A) Western blot of ASPA protein levels in wild-type and *cdc48-1* yeast strains expressing the indicated constructs cultured with

0.1 mM CuSO$_4$. Tubulin served as a loading control. (B) Western blot showing the levels of soluble (S) and insoluble (P) ASPA in wild-type and *cdc48-1* yeast strains expressing the indicated constructs in presence of 0.1 mM CuSO$_4$. Tubulin and Pma1 served as controls for the soluble and insoluble fractions, respectively. (C) Representative fluorescence microscopy images of GFP-ASPA C152W in a wild-type or *cdc48-1* strain. Scale bars are 4 μm. (D) Cells expressing GFP and containing aggregates were counted using fluorescence microscopy of wild-type (wt, C152W) and *cdc48-1* yeast strains expressing the indicated constructs. Data points indicate frequencies from independent experiments (minimum of 127 GFP-expressing cells, n = 6). The frequencies of cells containing aggregates were compared using a one-way ANOVA and Tukey's multiple comparisons test in GraphPad Prism. Means and standard deviations are shown. (E) Cells containing aggregates and the number of aggregates were counted by fluorescence microscopy of wild-type (wt, C152W) and *cdc48-1* yeast cells expressing the shown constructs. The frequencies of each observation in separate experiments are shown as data points (minimum of 127 GFP-expressing cells, n = 6). The aggregate patterns were compared using a two-way ANOVA and Tukey's multiple comparisons test. Means and standard deviations are shown.
(TIF)

**S6 Fig. ASPA degradation is ubiquitin-dependent.** U2OS cells transiently transfected to express HA-ubiquitin and a vector control, wild-type ASPA or C152W were treated with bortezomib (BZ, +) or as a control DMSO (-) followed by denaturing HA immunoprecipitation. Vinculin serves as a loading control. Note that the level of ubiquitinated ASPA C152W increases following treatment with bortezomib.
(TIF)

**S7 Fig. Knock-down of HSPA4 results in increased levels of insoluble ASPA.** (A) Knock-down of *HSPA4* in U2OS cells transiently transfected with the indicated constructs results in stabilization of ASPA. Successful knock-down was confirmed by blotting for HSPA4. Vinculin served as a loading control. (B) Cells were treated as in A, followed by differential centrifugation to separate soluble and insoluble protein into supernatant (S) and pellet (P) fractions. Successful knock-down was confirmed by blotting for HSPA4. GAPDH and the Na/K ATPase served as controls for soluble and insoluble proteins, respectively. Note that ASPA C152W is exclusively found in the insoluble fraction.
(TIF)

**S8 Fig. Hsp110-mediated ASPA degradation.** (A) U2OS cells transiently transfected to express ASPA C152W and HSPH1 or a vector control with and without *HSPH1* knock-down. Note that expression of episomal siRNA-resistant myc-tagged HSPH1 restores degradation of ASPA C152W upon knock-down of endogenous *HSPH1*. Vinculin served as a loading control. (B) Knock-down of HSPH1 expression does not lead to detectable changes in the level of Hsp70 or ubiquitin-protein conjugates. GAPDH served as a loading control.
(TIF)

**S1 Dataset. Numerical data.** A spreadsheet of the presented numerical data.
(XLSX)

## Acknowledgments

The authors thank Dr. Sofie V. Nielsen for helpful discussions of experimental procedures and comments on the manuscript, Anne-Marie Lauridsen and Søren Lindemose for excellent technical assistance, Dr. Tommer Ravid, Dr. Yves Barral, Dr. Kuninori Suzuku, Dr. Mads Gyrd-

Hansen, and Dr. Michael Lisby for sharing valuable reagents, and Dr. Frederic Rousseau and Dr. Joost Schymkowitz (VIB) for sharing the Limbo Software.

## Author Contributions

**Conceptualization:** Sarah K. Gersing, Claes Andréasson, Amelie Stein, Kresten Lindorff-Larsen, Rasmus Hartmann-Petersen.

**Formal analysis:** Sarah K. Gersing, Yong Wang, Martin Willemoës, Claes Andréasson, Amelie Stein, Kresten Lindorff-Larsen, Rasmus Hartmann-Petersen.

**Investigation:** Sarah K. Gersing, Yong Wang, Martin Grønbæk-Thygesen, Caroline Kampmeyer, Lene Clausen, Martin Willemoës.

**Methodology:** Claes Andréasson, Kresten Lindorff-Larsen.

**Project administration:** Rasmus Hartmann-Petersen.

**Supervision:** Kresten Lindorff-Larsen, Rasmus Hartmann-Petersen.

**Writing – original draft:** Sarah K. Gersing.

**Writing – review & editing:** Sarah K. Gersing, Yong Wang, Martin Grønbæk-Thygesen, Caroline Kampmeyer, Lene Clausen, Martin Willemoës, Claes Andréasson, Amelie Stein, Kresten Lindorff-Larsen, Rasmus Hartmann-Petersen.

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
