## [Decision Letter · Decision Letter 0]

25 Nov 2020

Dear Dr Hartmann-Petersen,

Thank you very much for submitting your Research Article entitled 'Evolutionarily conserved chaperone-mediated proteasomal degradation of a disease-linked aspartoacylase variant' to PLOS Genetics. Your manuscript was fully evaluated at the editorial level and by independent peer reviewers. The reviewers appreciated the attention to an important problem, but raised some substantial concerns about the current manuscript. Based on the reviews, we will not be able to accept this version of the manuscript, but we would be willing to review again a much-revised version. We cannot, of course, promise publication at that time.

If you decide to revise the manuscript for further consideration at PLOS Genetics, please aim to resubmit within the next 60 days, unless it will take extra time to address the concerns of the reviewers, in which case we would appreciate an expected resubmission date by email to plosgenetics@plos.org.

[LINK]

We are sorry that we cannot be more positive about your manuscript at this stage. Please do not hesitate to contact us if you have any concerns or questions.

Yours sincerely,

Hua Tang

Section Editor: Natural Variation

PLOS Genetics

Scott Williams

Section Editor: Natural Variation

PLOS Genetics

Reviewer's Responses to Questions

**Comments to the Authors:**

Reviewer #1: In the manuscript Gersing et al. describe their work on the cellular fate of an aspartoacylase (ASPA) variant that causes a fatal neurodegenerative disorder called Canavan disease. They demonstrate that the loss of function phenotype of the C152W ASPA variant is a result of an increased proteasome dependent degradation of the protein and characterize crucial components of the cellular protein quality control (PQC) system involved.

The manuscript is clearly written, well structured and technically sound. Relevant literature is discussed appropriately. Experimental data are of high quality and thorough description of the methods permits reproducibility. Conclusions are fully supported by the experimental data.

In contrast to a previous study, reporting that thermal and conformational stability of the C152W ASPA is similar to the wt-protein in vitro, the authors employ an in vivo approach, revealing reduced steady state levels of C152W. The findings imply a prominent role of the cellular PQC system in the loss of function phenotype of ASPA C152W. Accordingly, a possible involvement of PQC-components in the fate of ASPA wt and C152W was tested in different cellular backgrounds, resulting in the identification of Hsp110/HSPH1 as a crucial factor mediating proteasome dependent degradation of ASPA.

Based on their data, the authors are able to propose a model describing the interplay of PQC components and the proteasome system in ASPA degradation and discuss HSPH1 as a potential therapeutic target.

The findings presented by Gersing et al. provide new important insights into PQC and its link to protein degradation, which are crucial to understand the cellular mechanisms involved in the development of Canavan disease and other conditions with similar etiology.

Reviewer #2: The manuscript from Gersing et al. explores the mechanism of pathogenicity of the canavan disease-linked C152W variant of aspartoacylase (ASPA). The authors show convincing evidence that ASPA degradation dynamics is mediated by interaction with Hsp70 followed by ubiquitylation from ubr1, and finally proteasomal degradation mediated by Hsp110 in the yeast model. There is strong evidence that C152W is more aggregation prone than WT and therefore results in increased targeting for degradation. It is demonstrated that the disaggregase, Hsp104, somewhat rescues this aggregation phenotype but appears to accelerate degradation when expressed in human cells. Computational modeling presents convincing evidence that decreased steady state of the C152W variant is likely due to the exposure of a degron sequence, and follow-up reporter-based experiments confirm this hypothesis. Finally, the authors demonstrate that this degradation mechanism is conserved in human cells. The authors propose that targeting protein chaperones could therapeutically treat this degradation pathology.

Major Revisions:

1. An essential experiment to substantiate the claims of this paper is an enzyme activity assay demonstrating that the rate of catalysis in the NAA to acetate + aspartate is not significantly perturbed due to the C152W mutation. Although the references cited show from crystal structures that the C152W mutation is outside the active site, in order to support the claim and investigation that C152W pathogenic mechanism is exclusively due to increased degradation, evidence of equivalent enzymatic activity is needed.

2. The aggregation experiments in figure 2 are not entirely convincing though providing additional data analysis may be sufficient and not necessarily a repeat of the experiment. In addition to the quantity of aggregates, the distribution, area, and intensity of the foci and diffuse protein should also be measured. From the representative IHC, it appears that WT is more diffuse throughout the cells than appearing in aggregates. This feature can be quantified and calculated as a ratio of diffuse:aggregate signal, and this should demonstrate that the majority of ASPA C152W appears in foci in contrast to diffuse through the cytoplasm.

a. In addition, panel 2F and 2G can be merged by including 0 aggregates and 1 aggregate as data points.

b. Adding panel S1C to figure panel 2 would be desirable.

3. The degron experiment in Figure 3 C and D isn’t that helpful in supporting the claim that C152W exposes the degron during de novo folding. Since WT and C152W show equivalent results, it simply supports the claim that the sequence in and of itself contains a degron. But it does not show that C152W increases the accessibility/exposure of the degron; there needs to be a contrast. It might be useful to attempt an experiment eliminating the degron from ASPA or making sufficient amino acid substitutions to show that removing the sequence as a whole has a different effect than the C152W substitution.

a. Otherwise, consider relocating panel S2 to figure panel 3 and placing 3D in the supplement.

4. The claim that HSP’s and PQC machinery be targeted for therapeutic treatment of the disease is rather bold. Although the authors concede that these protein networks have broad involvement among the degradation dynamics of many proteins, evidence that targeting Hsp110 and Hsp70 does not substantially affect other targets is necessary to put forth such a claim. The authors can instead rephrase these statements as a call to further investigation into the possibility of therapeutically targeting these proteins to treat disease.

a. In relation to this point, some description of gene replacement and enzyme replacement therapies for Canavan Disease should be provided in greater detail. The authors could explain that this research focuses on to addressing misregulated degradation dynamics. (i.e. explain that this research relates to addressing protein misfolding diseases as a whole and Canavan Disease simply provides a useful context for this research). It would be helpful to mention other protein misfolding diseases that this research can be applied to. However, there is no definitive evidence among this research that Hsp70 and Hsp110 are indisputable therapeutic targets and therefore such a claim needs to be minimized.

b. The most useful conclusions from this research are that subtle rate changes in de novo folding can expose degrons during an intermediate state which subsequently accelerates degradation depleting essential metabolic enzymes. This in and of itself can result in a disease phenotype. Therefore, the focus of the manuscript ought to underscore the importance of investigating protein folding, degradation dynamics, and deconvolution of the pathway to degradation (Not the disease).

Minor Revisions:

1. Recommendation to modify title to reflect a conclusion of the research as opposed to a general topic. (e.g. Increased proteasomal degradation of disease-linked aspartoacylase variant is a major pathogenic mechanism of disease progression)

2. Recommendation to better clarify the difference between ASPA’s native folding and structure vs. the process of de novo folding. For example, in the abstract, “…indicating that the native state is structurally preserved” and “…prevents ASPA from reaching its stable native conformation” seem conflicting upon first read. Perhaps clarify the latter sentence to: “C152W substitution interferes with the de novo folding process resulting in increased proteasomal degradation before reaching its stable conformation”.

3. The sentence starting on line 155 can be rewritten as a comparison between C152W and WT. (e.g. “While WT ASPA was mainly soluble, the C152W variant, in contrast, demonstrated increased insolubility”)

4. The figure legend of 3A and related text should include the software used to produce the figures despite describing in the methods section.

5. I didn’t quite follow the logic in jumping immediately into studying HSP70. Perhaps substantiating the particular decision to investigate Hsp70 would be helpful to the text narrative. What other proteins were considered for analysis and why was Hsp70 chosen for careful investigation?

6. The illustration in figure 4A could be more detailed to help with a more general readership, e.g. labeling the proteasome.

7. The text regarding Cdc48 is a bit unclear. Perhaps including Cdc48’s mechanism in the illustration of panel 5A can better describe the involvement in the overall degradation pathway. It should also be clarified as to how Cdc48 relates to Hsp104.

8. At times it seems the word “stability” is conflated with “reduced aggregation/degradation propensity”. Please clarify the difference between protein folding (stability) and degradation dynamics (rate).

**Have all data underlying the figures and results presented in the manuscript been provided?**

Reviewer #1: Yes

Reviewer #2: Yes

PLOS authors have the option to publish the peer review history of their article (what does this mean?). If published, this will include your full peer review and any attached files.

Reviewer #1: No

Reviewer #2: No

---

## [Decision Letter · Decision Letter 1]

6 Apr 2021

Dear Dr Hartmann-Petersen,

We are pleased to inform you that your manuscript entitled "Mapping the degradation pathway of a disease-linked aspartoacylase variant" has been editorially accepted for publication in PLOS Genetics. Congratulations!

However, *please note that this is a conditional acceptance*: you must address the comments from Reviewer 2 during the pre-production process for the manuscript to be approved for final publication.

Yours sincerely,

Hua Tang

Section Editor: Natural Variation

PLOS Genetics

Scott Williams

Section Editor: Natural Variation

PLOS Genetics

Comments from the reviewers (if applicable):

Reviewer's Responses to Questions

**Comments to the Authors:**

Reviewer #1: In their revised manuscript the authors have sufficiently addressed the constructive points of criticism made by Reviewer 2.

As already stated in my review of the first submission, I think that the study provides valuable information to understand the biological phenotype of mutations that alter protein structure and their potential to cause disease. Therefore, the study is of high importance not only to the field of PQC, but also to Genetics and Genomics.

Reviewer #2: The authors are to be commended for comprehensively addressing all of my previous concerns, and I am happy to see these improvements incorporated into this manuscript. It is unfortunate the enzymatic activity attempts were unsuccessful; nevertheless, I am impressed and appreciate the rigorous effort to do so, and I think the attempt adds to the quality of the manuscript. The tone and flow of the manuscript is much improved, and I think it is successful at invoking interest and further discussion (relating to Major Rev. 4 and Minor Revisions). I appreciate the response from the authors regarding Major Rev. 2 and 3, and I think their decisions are appropriate. I identify only a couple more things that I would recommend be polished before publication below.

1. Regarding the diagrams in 5A and 7. It would be helpful to add a little more detail about the sequence in which Ubr1, Hsp104, Hsp110, and Hsp70 interact and direct unfolded proteins to certain fates. Furthermore, figure 5A and 7, along with the text and evidence in lines 269-301 (of the manuscript with track changes) are difficult to reconcile. It appears to be the case that Hsp104 is a completely separate mechanism than the Hsp70 – Hsp110 mechanism, and that its sole purpose is to resolubilize aggregated ASPA. Therefore, its mechanism is not coupled with proteasome degradation. Can you please better articulate the sequence of events?

2. Unless I am missing something, in line 319 the word “stabilization” appears to be conflated with degradation dynamics. The protein in these experiments is identified exclusively in the insoluble fraction. This should not be interpreted as stability as the term is accompanied by the assumption that the protein is soluble and functional. (Decreased catalytic activity is likely also in part due to aggregation). Therefore, I think this phrasing is misleading for readers. Statements like this are better termed as “increased accumulation” and/or “attenuated degradation”.

a. Another instance that should be rephased is lines 254-256

3. As a follow-up to previous Major Revision #1, I noticed in your citations that their enzyme activity experiments relied on cell lysates and not purified protein as is attempted here. It might be good to mention why the latter approaches were attempted and emphasize the difficulties in reconciling this evidence.

4. It would be helpful to clarify among the figures when the Cu promoter is on or off (i.e. Figure 1B,C) and add this detail to the graphic in Figure 1A.

**Have all data underlying the figures and results presented in the manuscript been provided?**

Reviewer #1: Yes

Reviewer #2: Yes

PLOS authors have the option to publish the peer review history of their article (what does this mean?). If published, this will include your full peer review and any attached files.

Reviewer #1: No

Reviewer #2: No

**Data Deposition**

http://datadryad.org/submit?journalID=pgenetics&manu=PGENETICS-D-20-01411R1

**Press Queries**

---

## [Editor Report · Acceptance letter]

15 Apr 2021

PGENETICS-D-20-01411R1 

Mapping the degradation pathway of a disease-linked aspartoacylase variant 

Dear Dr Hartmann-Petersen, 

We are pleased to inform you that your manuscript entitled "Mapping the degradation pathway of a disease-linked aspartoacylase variant" has been formally accepted for publication in PLOS Genetics! Your manuscript is now with our production department and you will be notified of the publication date in due course.

With kind regards,

Katalin Szabo

PLOS Genetics

On behalf of:
